# Uncertainty-Aware Surrogate-based Amortized Bayesian Inference for Computationally Expensive Models

**Stefania Scheurer**[*]  *stefania.scheurer@iws.uni-stuttgart.de*
*Department of Stochastic Simulation and Safety Research for Hydrosystems*
*Cluster of Excellence SimTech*
*University of Stuttgart*

**Philipp Reiser**[*]  *philipp.reiser@simtech.uni-stuttgart.de*
*Cluster of Excellence SimTech*
*University of Stuttgart*

**Tim Brünnette**  *tim.bruennette@iws.uni-stuttgart.de*
*Department of Stochastic Simulation and Safety Research for Hydrosystems*
*University of Stuttgart*

**Wolfgang Nowak**  *wolfgang.nowak@iws.uni-stuttgart.de*
*Department of Stochastic Simulation and Safety Research for Hydrosystems*
*Cluster of Excellence SimTech*
*University of Stuttgart*

**Anneli Guthke**  *anneli.guthke@simtech.uni-stuttgart.de*
*Cluster of Excellence SimTech*
*University of Stuttgart*

**Paul-Christian Bürkner**  *paul.buerkner@tu-dortmund.de*
*Department of Statistics*
*TU Dortmund University*

**Reviewed on OpenReview:** *https://openreview.net/forum?id=aVSoQXbfy1*

## Abstract

Bayesian inference typically relies on a large number of model evaluations to estimate posterior distributions. Established methods like Markov Chain Monte Carlo (MCMC) and Amortized Bayesian Inference (ABI) can become computationally challenging. While ABI enables fast inference *after* training, generating sufficient training data still requires thousands of model simulations, which is infeasible for expensive models. Surrogate models offer a solution by providing *approximate* simulations at a lower computational cost, allowing the generation of large datasets for training. However, the introduced approximation errors and uncertainties can lead to overconfident posterior estimates. To address this, we propose Uncertainty-Aware Surrogate-based Amortized Bayesian Inference (UA-SABI) – a framework that combines surrogate modeling and ABI while explicitly quantifying and propagating surrogate uncertainties through the inference pipeline. Our experiments show that this approach enables reliable, fast, and repeated Bayesian inference for computationally expensive models, even under tight time constraints.

---

[*]These authors contributed equally to this work.

# 1 Introduction

Mathematical models are essential for simulating real-world processes, typically mapping parameters to observable data. Estimating these parameters from real-world observations, i.e., the inverse problem, is fundamental across scientific disciplines (e.g. Etz & Vandekerckhove, 2018; Von Toussaint, 2011; Hülsenbeck et al., 2001; Ellison, 2004). However, because models never perfectly capture reality and observed data are often sparse and imprecise, parameter estimation inherently involves uncertainty. Bayesian inference offers a systematic framework for estimating parameters while incorporating uncertainty in a statistically grounded manner (e.g. Gelman et al., 1995).

Markov Chain Monte Carlo (MCMC) methods are widely used for Bayesian inference to generate high-quality samples from the posterior distribution given fixed observations (Gilks et al., 1995). However, MCMC is computationally expensive and slow, rendering it impractical in scenarios where near-instant inference is required (Robert et al., 2018). Near-instant inference is, for example, necessary in adaptive robotic control or closed-loop medical devices, where parameters such as object mass or patient sensitivity must be estimated immediately to allow accurate prediction and safe action (e.g. Marlier, 2024; Tasoujian et al., 2020; Malagutti et al., 2023). Furthermore, each new set of observations requires restarting the entire process, further increasing costs when inference is needed for multiple datasets. Multiple inference runs may be needed for ongoing adaptation as conditions evolve or more data become available. Tracking the spread of a disease such as influenza or COVID-19 is one example of this. New case data continuously arrive, and separate datasets exist for different regions, requiring repeated Bayesian updates to infer transmission rates or reproduction numbers for each location (e.g. Radev et al., 2021; Yang et al., 2015). MCMC also relies on a known likelihood function that can be evaluated either analytically or numerically.

Amortized Bayesian Inference (ABI), a deep-learning-based approach originating from simulation-based inference (SBI), addresses these limitations (Cranmer et al., 2020; Radev et al., 2020; Lückmann et al., 2021). SBI methods are typically employed when evaluating the likelihood is infeasible, but simulations from the model are possible, allowing learning a mapping from observed data to the posterior of the model parameters via simulated data. ABI in particular leverages generative neural networks to learn this mapping. Training data is generated through multiple model evaluations. Once trained, ABI enables near-instant posterior inference for new datasets, as the computational cost is incurred during training. Being likelihood-free makes ABI well-suited for complex problems involving noisy or high-dimensional data, where evaluating the likelihood is infeasible or unreliable. However, the learned posterior only approximates the true posterior, and high accuracy requires well-designed network architectures and extensive training data.

Both MCMC and ABI face limitations when the simulation model is computationally expensive, since both require a large number of model evaluations. MCMC requires many likelihood evaluations per generated posterior sample; ABI requires extensive model simulations to generate sufficient amounts of training data. As a result, if computational time is limited, obtaining a good posterior becomes infeasible for both approaches. Note that such computational effort can arise for different reasons – for instance, from a high-dimensional parameter space or from the complexity of the governing equations. In Earth sciences, for example, low-dimensional models can be computationally very expensive, particularly if they involve (coupled systems of) non-linear partial differential equations or long simulated time periods (e.g. Mohammadi et al., 2018; Hommel et al., 2015), resulting in runtimes up to multiple days for a single model run.

Our goal is to enable ABI for computationally expensive models to benefit from the aforementioned advantages that ABI offers. To this end, surrogate models are the crucial tool, allowing us to reduce the computational cost of expensive simulations for training data generation. Surrogate models (Sudret, 2008; Rasmussen & Williams, 2006), however, are only approximations (i.e., imperfect representations) of the reference simulation, and therefore not necessarily reliable. It is crucial to quantify the uncertainties associated with surrogate modeling, such as those arising from limited training data or any inherent inflexibility of the surrogate. These uncertainties then need to be propagated through the inference pipeline, ensuring consistent posterior estimation. Previous methods have integrated these uncertainties into MCMC methods with surrogate models (e.g. Lückmann et al., 2018; Zhang et al., 2020; Reiser et al., 2025). However, this requires many additional MCMC runs to incorporate the surrogate uncertainty, largely eliminating the computational advantages of surrogates.

We introduce Uncertainty-Aware Surrogate-based Amortized Bayesian Inference (UA-SABI), which enables ABI for computationally expensive models while accounting for the uncertainty inherent in surrogate approximations. By leveraging a surrogate, we reduce the training error that arises from insufficient data during ABI training, as simulating from the surrogate is easy and fast. This comes at the cost of introducing a surrogate approximation error. However, unlike the ABI training error, the surrogate approximation error can be quantified and propagated, enabling reliable ABI for computationally expensive models.

For this purpose, we adapt and further develop the uncertainty estimation and propagation method for surrogate modeling proposed in Reiser et al. (2025). Our main novelty is to replace the MCMC with ABI for inference. This novelty and its benefits can be viewed from two perspectives: 1) Replacing MCMC with ABI substantially improves sampling efficiency, as parallel chains for draws from the surrogate posterior are no longer required. 2) UA-SABI enables ABI training with surrogate data while accounting for the uncertainty inherent in the surrogate approximation. This significantly reduces the required runs from the expensive simulation model while retaining the amortization property and, therefore, enabling ABI for computationally expensive models.

An illustration of the workflow, including training phases and inference, is given in Fig. 1. We demonstrate the workflow and substantiate the claimed advantages both theoretically and practically with a toy example and two real-world case studies.

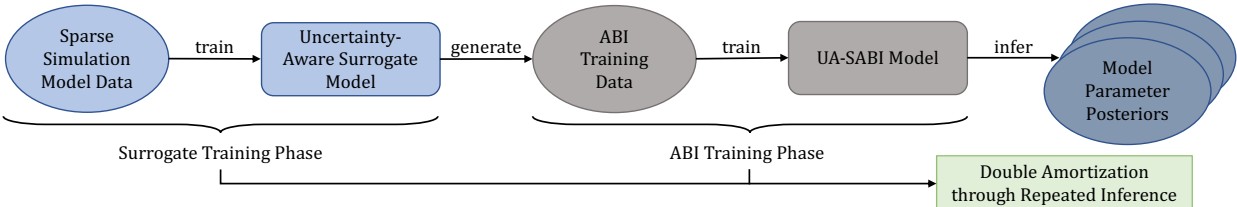

Figure 1: Illustrative workflow of UA-SABI training and inference.

## 2 Methods

### 2.1 Amortized Bayesian Inference (ABI)

This work focuses on using ABI to infer the parameters of computationally expensive simulation models. The typical ABI procedure (Radev et al., 2020) can be divided into a training phase and an inference phase. Training includes simulating data and then training a neural posterior estimator (NPE) on that data (Papamakarios & Murray, 2016; Durkan et al., 2019; Greenberg et al., 2019b).

**ABI Training Data Generation**  Training data is generated using a simulation model $M = M(\mathbf{x}, \boldsymbol{\omega})$, typically a realistic first-principles model. $M$ depends on inputs $\mathbf{x}$ and parameters $\boldsymbol{\omega}$ and yields an output $\mathbf{y}$. Additionally, it can contain stochastic parts omitted in the following to simplify the notation. To learn the mapping from observations to posterior draws, a training set $D_B = \{(\mathbf{x}^{(i)}, \boldsymbol{\omega}^{(i)}, \mathbf{y}^{(i)})\}_{i=1}^{N_B}$ of $N_B$ samples must first be generated. This can be performed by simulating $N_B$ pairs of parameters, inputs, and outputs from the simulation model:

$$\mathbf{y}^{(i)} = M(\mathbf{x}^{(i)}, \boldsymbol{\omega}^{(i)}) \quad \text{with} \quad (\mathbf{x}^{(i)}, \boldsymbol{\omega}^{(i)}) \sim p(\mathbf{x}, \boldsymbol{\omega}). \tag{1}$$

**Neural Posterior Estimation**  After creating the training dataset, the NPE, including a summary network $S_{\boldsymbol{\theta}}(\mathbf{x}, \mathbf{y})$ and an inference network $I_{\boldsymbol{\varphi}}(\mathbf{s})$, must be trained. The goal of $S_{\boldsymbol{\theta}}(\mathbf{x}, \mathbf{y})$ is to embed (stacked) observations, i.e., model inputs and outputs $(\mathbf{x}, \mathbf{y})$ of potentially variable length (e.g., a measurement series) in a fixed-length vector $\mathbf{s}$ that serves as input to $I_{\boldsymbol{\varphi}}(\mathbf{s})$. Then, $I_{\boldsymbol{\varphi}}(\mathbf{s})$ generates samples from the approximate posterior $q_{\boldsymbol{\varphi}}(\boldsymbol{\omega} \mid \mathbf{s})$. Here, $\boldsymbol{\theta}$ and $\boldsymbol{\varphi}$ are learnable coefficients in $S_{\boldsymbol{\theta}}(\mathbf{x}, \mathbf{y})$ and $I_{\boldsymbol{\varphi}}(\mathbf{s})$. In our case studies, we use coupling flows (Papamakarios et al., 2021) as inference networks. Coupling flows have been empirically and theoretically shown to be expressive and allow for fast evaluation and sampling (Draxler et al., 2025).

However, ABI is generally flexible in its choice for the inference network and works with any generative neural network (e.g. Ardizzone et al., 2019; Padmanabha & Zabaras, 2021; Denker et al., 2021; Wildberger et al., 2023; Schmitt et al., 2024b). Thus, the summary and inference networks are modular and can be independently replaced or adapted to suit the specific problem at hand.

The optimal parameters of the summary and inference network $(\boldsymbol{\theta}^*, \boldsymbol{\varphi}^*)$ are obtained jointly by minimizing the Kullback-Leibler (KL) divergence between the true posterior $p(\boldsymbol{\omega} \mid \mathbf{x}, \mathbf{y})$ and the approximate posterior $q_{\boldsymbol{\varphi}}(\boldsymbol{\omega} \mid \mathbf{s})$ (Radev et al., 2020):

$$(\boldsymbol{\theta}^*, \boldsymbol{\varphi}^*) = \underset{\boldsymbol{\theta}, \boldsymbol{\varphi}}{\operatorname{argmin}} \, \mathbb{E}_{p(\mathbf{x}, \mathbf{y})}[\mathrm{KL}(p(\boldsymbol{\omega} \mid \mathbf{x}, \mathbf{y}) \, || \, q_{\boldsymbol{\varphi}}(\boldsymbol{\omega} \mid \mathbf{s}))] \tag{2}$$

$$= \underset{\boldsymbol{\theta}, \boldsymbol{\varphi}}{\operatorname{argmin}} \, \mathbb{E}_{p(\mathbf{x}, \mathbf{y})}[\mathbb{E}_{p(\boldsymbol{\omega}|\mathbf{x},\mathbf{y})}[-\log q_{\boldsymbol{\varphi}}(\boldsymbol{\omega} \mid \mathbf{s})]] \tag{3}$$

$$= \underset{\boldsymbol{\theta}, \boldsymbol{\varphi}}{\operatorname{argmin}} \, - \iiint p(\mathbf{x}, \mathbf{y}, \boldsymbol{\omega}) \log q_{\boldsymbol{\varphi}}(\boldsymbol{\omega} \mid S_{\boldsymbol{\theta}}(\mathbf{x}, \mathbf{y})) \, \mathrm{d}\mathbf{x} \, \mathrm{d}\mathbf{y} \, \mathrm{d}\boldsymbol{\omega}. \tag{4}$$

According to Radev et al. (2020), the expectations can be approximated with the Monte Carlo estimates, utilizing simulations $\{(\mathbf{x}^{(i)}, \boldsymbol{\omega}^{(i)}, \mathbf{y}^{(i)})\}_{i=1}^{N_B}$ from the (potentially expensive) simulation model (1). This results in the following loss function:

$$L(\boldsymbol{\varphi}, \boldsymbol{\theta}) = - \sum_{i=1}^{N_B} \log q_{\boldsymbol{\varphi}}(\boldsymbol{\omega}^{(i)} \mid S_{\boldsymbol{\theta}}(\mathbf{x}^{(i)}, \mathbf{y}^{(i)})). \tag{5}$$

## 2.2 Uncertainty-Aware Surrogate Modeling

Uncertainty-aware surrogate modeling is introduced in and adapted from Reiser et al. (2025). A surrogate model $\widetilde{M}$ aims to approximate the behavior of a simulation model $M$ with negligible computational cost. The surrogate model is typically parametrized by learnable surrogate coefficients, denoted as $\mathbf{c}$. The simulation model output can then be approximated by

$$\mathbf{y} = M(\mathbf{x}, \boldsymbol{\omega}) \approx \widetilde{M}_{\mathbf{c}}(\mathbf{x}, \boldsymbol{\omega}) = \widetilde{\mathbf{y}}. \tag{6}$$

However, due to limited simulation data from $M$ used for training $\widetilde{M}_{\mathbf{c}}$, the surrogate coefficients $\mathbf{c}$ can only be estimated with substantial epistemic uncertainty. For the same reason, the expressibility of $\widetilde{M}_{\mathbf{c}}$ must be restricted, i.e., the complexity of $\widetilde{M}_{\mathbf{c}}$ must be kept relatively low. This introduces an approximation error, denoted as $\epsilon$, which captures the misspecification of the surrogate. In practice, one usually has to expect a non-negligible approximation error due to limitations in model capacity, optimization challenges, or regularization constraints. We assume the approximation error $\epsilon \sim p(\epsilon \mid \sigma)$ to follow a distribution parametrized by $\sigma$, leading to the error-adjusted surrogate output

$$\widetilde{\mathbf{y}}_{\epsilon} = f(\widetilde{\mathbf{y}}, \epsilon) \quad \text{with} \quad \epsilon \sim p(\epsilon \mid \sigma), \tag{7}$$

which is essentially a perturbed version of $\widetilde{\mathbf{y}}$. This setup defines the surrogate likelihood

$$p(\widetilde{\mathbf{y}}_{\epsilon} \mid \widetilde{\mathbf{y}}, \sigma) = p(\widetilde{\mathbf{y}}_{\epsilon} \mid \mathbf{x}, \boldsymbol{\omega}, \mathbf{c}, \sigma), \tag{8}$$

that approximates the (usually unknown) likelihood of the simulation model.

Using **sparse** training data ($N_T \ll N_B$) obtained from the simulation model $D_T = \{(\mathbf{x}^{(i)}, \boldsymbol{\omega}^{(i)}, \mathbf{y}^{(i)})\}_{i=1}^{N_T}$, with $\mathbf{y}^{(i)} = M(\mathbf{x}^{(i)}, \boldsymbol{\omega}^{(i)})$ and $(\mathbf{x}^{(i)}, \boldsymbol{\omega}^{(i)})$ sampled from a given prior, we infer the surrogate model coefficients $\mathbf{c}$ and $\sigma$ via Bayesian inference. This enables capturing both the surrogate model's epistemic uncertainty and the irreducible approximation error. For this purpose, we consider the likelihood equation 8 and the prior $p(\mathbf{c}, \sigma)$. The joint posterior over the surrogate parameters and the approximation noise, given training data $D_T$ from the simulation model $M$, is then

$$p(\mathbf{c}, \sigma \mid D_T) \propto \prod_{i=1}^{N_T} p(\mathbf{y}^{(i)} \mid \mathbf{x}^{(i)}, \boldsymbol{\omega}^{(i)}, \mathbf{c}, \sigma) \, p(\mathbf{c}, \sigma), \tag{9}$$

assuming an i.i.d. sampled approximation error. This assumption serves as a simple stand-in, since surrogate construction is not the focus of this work. It suffices to demonstrate that even a basic error model outperforms making no error assumption at all. In general, UA-SABI is not limited to the i.i.d. case; heteroscedastic surrogate error models (e.g. Kohlhaas et al., 2023) are fully compatible with the method.

The conditions under which $\widetilde{M}$ is considered optimal are implied by the assumed distributions of $\mathbf{x}$ and $\boldsymbol{\omega}$ in $D_T$ and the specified error model $p(\epsilon \mid \sigma)$. Sampling-based algorithms such as MCMC can be used to estimate the surrogate posterior equation 9. For the considered surrogates, this is tractable since their likelihoods are fast to evaluate, and the surrogate posterior – compared to model parameter posteriors across datasets – is estimated only once. While this formulation fully captures surrogate uncertainty, we now temporarily simplify the setup by omitting this uncertainty to introduce an uncertainty-*unaware* baseline method – Surrogate-based ABI. This allows us to isolate and demonstrate the benefit of propagating surrogate uncertainty through to inference.

### 2.3 Surrogate-based ABI (SABI)

The quality of the NPE depends on a sufficient training budget $D_B$ to ensure that $p(\boldsymbol{\omega} \mid \mathbf{x}, \mathbf{y})$ is well approximated via $q_{\boldsymbol{\varphi}}(\boldsymbol{\omega} \mid S_{\boldsymbol{\theta}}(\mathbf{x}, \mathbf{y}))$. However, for the expensive models considered here, generating sufficient ABI training data becomes infeasible. To mitigate this, we propose using a surrogate model of the expensive simulator to efficiently generate sufficient training data. Surrogate-based ABI (SABI) thus differs from standard ABI only in the generation of training data. Instead of using the expensive model, a (point) surrogate model with a point estimate $\overline{\mathbf{c}}$ is used to generate the training data $D_B = \{(\mathbf{x}^{(i)}, \boldsymbol{\omega}^{(i)}, \widetilde{\mathbf{y}}^{(i)})\}_{i=1}^{N_B}$:

$$\widetilde{\mathbf{y}}^{(i)} = \widetilde{M_{\overline{\mathbf{c}}}}(\mathbf{x}^{(i)}, \boldsymbol{\omega}^{(i)}) \quad \text{with} \quad (\mathbf{x}^{(i)}, \boldsymbol{\omega}^{(i)}) \sim p(\mathbf{x}, \boldsymbol{\omega}). \tag{10}$$

For the point surrogate, the median of the posterior in Eq. (9) can be used as it is robust to outliers and invariant under monotonic transformations, but also the mean is a valid option.

However, in general, $\widetilde{M_{\mathbf{c}}}$ is only an approximation of the simulation model $M$. Therefore, the SABI posterior trained with surrogate data $p(\boldsymbol{\omega} \mid \mathbf{x}, \widetilde{\mathbf{y}})$ will be systematically different from the true posterior $p(\boldsymbol{\omega} \mid \mathbf{x}, \mathbf{y})$, as no surrogate uncertainties or errors are considered in the ABI training process. Specifically, neither the epistemic uncertainty of $\widetilde{M_{\mathbf{c}}}$, nor its irreducible approximation error is propagated. As we show in our experiments, this can lead to highly inaccurate posterior approximations.

### 2.4 Uncertainty-Aware Surrogate-based ABI (UA-SABI)

To ensure reliable inference, the uncertainty of the surrogate model needs to be propagated through the ABI training and inference procedure, as illustrated in Fig. 1. This results in our proposed Uncertainty-Aware Surrogate-based Amortized Bayesian Inference (UA-SABI) approach.

Instead of generating training data with the point surrogate model, UA-SABI utilizes the uncertainty-aware surrogate model:

$$\widetilde{\mathbf{y}}_{\epsilon}^{(i)} \sim p(\widetilde{\mathbf{y}}_{\epsilon} \mid \mathbf{x}^{(i)}, \boldsymbol{\omega}^{(i)}, \mathbf{c}^{(i)}, \sigma^{(i)}) \quad \text{with} \quad (\mathbf{x}^{(i)}, \boldsymbol{\omega}^{(i)}) \sim p(\mathbf{x}, \boldsymbol{\omega}), (\mathbf{c}^{(i)}, \sigma^{(i)}) \sim p(\mathbf{c}, \sigma \mid D_T). \tag{11}$$

To propagate uncertainty to the ABI training, we now additionally draw the surrogate coefficients $\mathbf{c}^{(i)}$ and the approximation error parameter $\sigma^{(i)}$ from the surrogate posterior equation 9. These coefficients are then used to evaluate the surrogate model and obtain $\widetilde{\mathbf{y}}^{(i)}$. The pair $(\widetilde{\mathbf{y}}^{(i)}, \sigma^{(i)})$ is subsequently used to generate a sample $\widetilde{\mathbf{y}}_{\epsilon}^{(i)}$ from the distribution of the error-adjusted surrogate output. We now approximate the KL divergence in Eq. (5) as:

$$L(\boldsymbol{\varphi}, \boldsymbol{\theta}) = -\sum_{i=1}^{N_B} \log q_{\boldsymbol{\varphi}}(\boldsymbol{\omega}^{(i)} \mid S_{\boldsymbol{\theta}}(\mathbf{x}^{(i)}, \widetilde{\mathbf{y}}_{\epsilon}^{(i)})). \tag{12}$$

Figure 2 gives a detailed graphical overview. Additionally, a pseudocode summarizing the UA-SABI training procedure can be found in Appendix A.

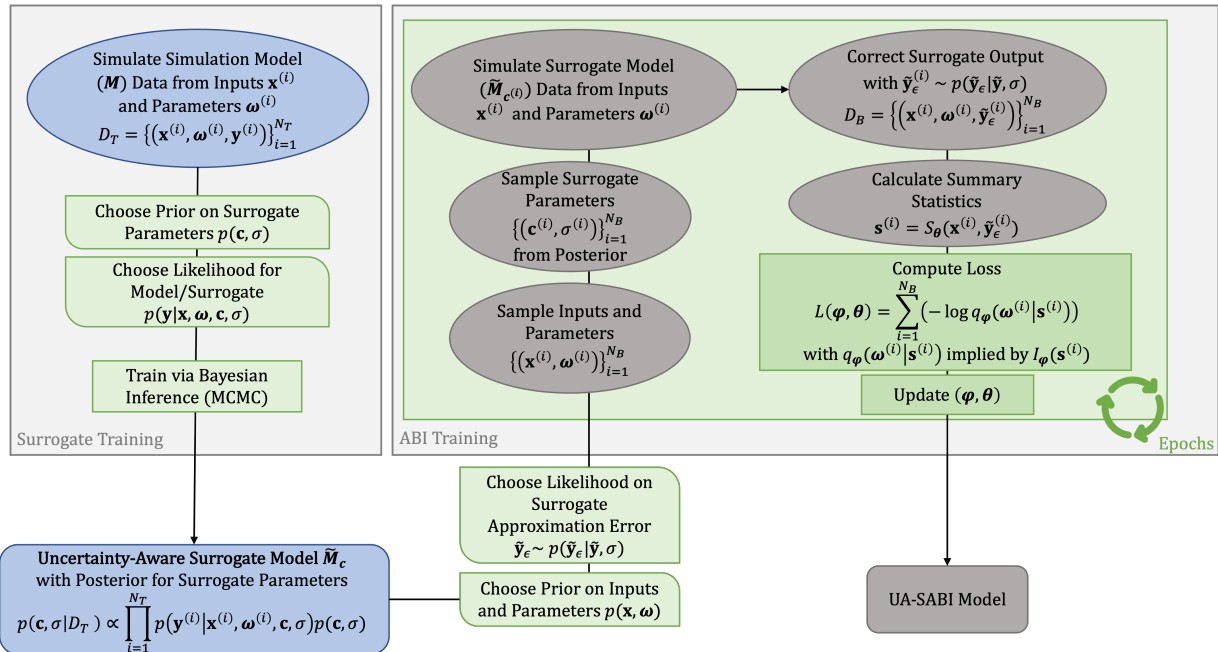

Figure 2: Detailed graphical overview of UA-SABI training as implemented in Algorithm 1. First phase: training of uncertainty-aware surrogate model using sparse simulation data, yielding a posterior over surrogate parameters. Second phase: ABI training using surrogate-generated data, where full surrogate uncertainty is propagated through sampled surrogate outputs.

## 3 Related Work

**Multi-Fidelity Approaches in SBI** Multi-fidelity methods leverage models or simulations of varying accuracy and cost to accelerate inference, combining cheap, approximate simulations with a smaller number of expensive, high-fidelity runs. In Approximate Bayesian Computation (ABC), such strategies reduce computational burden: Warne et al. (2018) apply multi-level Monte Carlo variance reduction to rejection sampling in ABC, improving efficiency. Prescott & Baker (2020) introduce a multi-fidelity ABC approach that combines early acceptance and rejection of parameter samples and selectively runs high-fidelity simulations based on low-fidelity outputs, balancing computational cost and accuracy. Building on this, Prescott & Baker (2021) integrate the multi-fidelity strategy with sequential Monte Carlo sampling, further enhancing efficiency by more effectively exploring the parameter space for MF-ABC proposals. More recent work by Warne et al. (2022) integrates low- and high-fidelity simulations within a multilevel Monte Carlo framework to accelerate SBI for partially observed stochastic processes, using inexpensive approximations to preselect parameters and high-fidelity simulations to refine posterior estimates, while Prescott et al. (2024) dynamically select between low- and high-fidelity models based on predictive accuracy, optimizing resource allocation. In multi-fidelity simulation-based Inference (MF-SBI) (Krouglova et al., 2025) a neural posterior estimator is first pre-trained on large amounts of low-fidelity data and then fine-tuned with a smaller set of high-fidelity simulations. While sharing similar goals, none of the aforementioned methods explicitly take the uncertainty of the low-fidelity simulations into account. UA-SABI's uncertainty-awareness guards against overconfidence in regions with scarce data – a protection which appears not to be in place for the low-fidelity training regions in MF-SBI, for example. Further, unlike many of the multi-fidelity methods, UA-SABI does not require domain expertise for the construction of the surrogate.

**Sequential Neural Posterior Estimation (S-NPE)**  S-NPE iteratively refines posterior estimates by adaptively choosing simulations that are most informative, improving efficiency. Papamakarios & Murray (2016) use initial simulations to fit a Bayesian conditional density estimator for the posterior and then sample parameters from the current posterior estimate to guide the next round of simulations. Similarly, Toni et al. (2009) introduced Sequential ABC (ABC-SMC), which iteratively refines posterior distributions by progressively tightening acceptance thresholds. Kulkarni & Moritz (2023) build on ABC-SMC, leveraging massively parallel architectures, such as GPUs, to more efficiently explore the parameter space. Lueckmann et al. (2017) sequentially train a Bayesian mixture-density network. Greenberg et al. (2019a) developed Automatic Posterior Transformation, which employs flexible flow-based density estimators and dynamically updated proposal distributions. Deistler et al. (2022) introduce truncated S-NPE, restricting simulations to truncated regions around high-probability areas. However, UA-SABI and S-NPE pursue related but distinct goals: UA-SABI focuses on fully amortized inference, enabling fast posterior estimation for new datasets through one-time training, whereas S-NPE sequentially refines the posterior on the same dataset to improve accuracy and sample efficiency across rounds. Additionally, the referenced S-NPE procedures implement an active learning strategy for simulation selection, assuming data are generated iteratively, while UA-SABI can also operate on pre-existing simulation datasets (offline training).

**Gaussian Process Surrogates in SBI**  Several methods employ Gaussian Process (GP) surrogates to improve sampling efficiency in likelihood-free inference. Meeds & Welling (2014) construct a GP to emulate the expensive simulator and use its predictive mean and uncertainty to estimate the acceptance probability in ABC, avoiding simulator calls when the surrogate is sufficiently confident. Wilkinson (2014) fit a GP to the ABC log-likelihood to prune implausible parameter regions before sampling, greatly reducing simulation cost. Bayesian Optimization likelihood-free inference (Gutmann & Corander, 2016) fits a GP to the discrepancy between simulated and observed data and uses Bayesian optimization to adaptively choose simulator parameters, concentrating effort on informative regions and enabling accurate posterior estimation with few simulations. As in UA-SABI, these approaches apply surrogates, focusing specifically on GPs. However, they employ the surrogate to intelligently select simulation parameters for subsequent likelihood-free inference, whereas UA-SABI uses the surrogate itself to directly generate training data and is agnostic to the surrogate model class.

**Structure-Aware SBI Methods**  Several computational strategies exploit the structure of simulators or information about the inference problem to improve efficiency in SBI. Compositional simulation-based inference for time series (Gloeckler et al., 2025) breaks long, high-dimensional time series into local transitions, performing inference on each step and composing the results to recover a global posterior. Cost-aware simulation-based inference (Bharti et al., 2025) tackles heterogeneous simulation costs by prioritizing cheaper and more informative simulations using a combination of rejection and importance sampling, introducing a cost function to guide allocation of simulation resources. In contrast to UA-SABI, structure-aware methods exploit knowledge about the simulator, whereas UA-SABI treats the simulator as a black box and requires no such knowledge.

**Neural Likelihood Estimation**  Neural likelihood estimation is a likelihood-free inference approach that models the likelihood function directly. Sequential neural likelihood (Papamakarios et al., 2019) uses autoregressive neural flows and focuses simulations on high-posterior regions for efficient inference. Bayesian synthetic likelihood (Price et al., 2018) approximates the likelihood of summary statistics with a multivariate normal, incorporating their uncertainty to provide robust and accurate inference. Jointly Amortized Neural Approximation (JANA) (Radev et al., 2023a) combines neural likelihood and posterior estimation. In addition to learning a neural posterior, it simultaneously learns a neural likelihood to enable marginal likelihood estimation, which is essential for tasks such as Bayesian model selection. However, its objective differs slightly, as it aims to learn both the likelihood, serving as a surrogate model, and the posterior jointly. Such a highly parameterized neural likelihood requires a sufficient amount of training data (Frazier et al., 2024). While UA-SABI also requires simulation data to train a surrogate, it achieves effective inference with far fewer simulations compared to neural likelihood estimation, making it more practical for very expensive models.

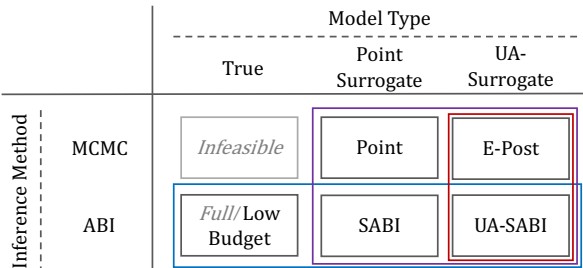

Figure 3: Context of UA-SABI and its alternatives under tight computational constraints.

**Low-Budget Simulation-based Inference with Bayesian Neural Networks** Delaunoy et al. (2024) also focus on the challenge of limited training data in ABI. They employ Bayesian Neural Networks (BNNs), which model uncertainty via distributions over network parameters. However, selecting an appropriate distribution and defining a meaningful prior for weights and biases remains speculative and can undermine posterior reliability, as noted in (Delaunoy et al., 2024). Moreover, BNNs still demand substantial training data and compute resources (~25,000 GPU hours in Delaunoy et al. (2024)), thus only shifting computational resources from evaluating expensive simulation models to training BNNs.

**Uncertainty-Aware Surrogate-Based Inference** Reiser et al. (2025) consider the same setting, replacing a computationally expensive model with an uncertainty-aware surrogate. However, their inference approach differs: they run MCMC separately for each observation set and for every surrogate sample. This results in a computational cost that scales with both the number of observations and the number of surrogate samples, making their method significantly more expensive. Their proposed method, E-Post, marginalizes over the surrogate coefficient posterior $p(\mathbf{c}, \sigma \mid D_T)$ to obtain the posterior of the parameters of interest $\boldsymbol{\omega}$:

$$p(\boldsymbol{\omega} \mid \mathbf{y}) = \iint p(\boldsymbol{\omega} \mid \mathbf{y}, \mathbf{c}, \sigma) \, p(\mathbf{c}, \sigma \mid D_T) \, \mathrm{d}\mathbf{c} \, \mathrm{d}\sigma. \tag{13}$$

In practice, this integral is approximated via a Monte Carlo integration with a sufficient number of samples from the surrogate posterior $p(\mathbf{c}, \sigma \mid D_T)$.

During inference with UA-SABI, this explicit marginalization is not required, as the NPE is trained over the entire posterior space of both the outputs and the surrogate parameters. This results in a form of double amortization of the surrogate and ABI training costs through repeated inference. Since UA-SABI and E-Post share the same overall objective and framework, E-Post with MCMC inference serves as a benchmark for UA-SABI. A formal proof of asymptotic equivalence is given in Proposition 1 in Appendix B.

## 4 Case Studies

We evaluate and benchmark UA-SABI against its uncertainty-unaware counterpart SABI, the corresponding MCMC-based methods, and (if possible) standard ABI using a toy example and two real-world case studies. Code is publicly available on GitHub[1].

### 4.1 Objectives

We aim to validate the parameter posterior obtained with UA-SABI by assessing its quality and justifying the computational effort required to train the surrogate by reducing the effort to train the ABI model. Our central research question is "Can we train ABI with surrogate data, do we have to, and, if so, how should we account for the uncertainty introduced by the surrogate?". To keep focus on that, we deliberately remove unrelated complexities. Specifically, in our real-world case studies, we use computationally expensive but low-dimensional problems. High-dimensional models present significant challenges for surrogate training, as they

---

[1] https://github.com/LS3-university-of-stuttgart/ua-sabi-paper

typically require substantially more data to adequately capture the model's behavior. Additionally, inference on high-dimensional problems becomes more complex due to multi-modality and non-identifiability issues that complicate posterior estimation. This would make result interpretation less transparent and potentially obscure insights about the proposed approach itself.

We want to show the advantage of using surrogate data instead of scarce simulation data, and demonstrate the importance of quantifying and propagating the uncertainty of the surrogate model. To do so, we compare UA-SABI, SABI, and standard ABI for the true model – trained with full and low budget (shown in blue in Fig. 3). To validate the posteriors produced by these methods, we also compare them with those obtained using the corresponding MCMC-based approach (shown in purple in Fig. 3). Further, to highlight the efficiency of our method, we compare the runtimes of UA-SABI and E-Post (shown in red in Fig. 3).

To assess posterior calibration for our first two case studies, we generate multiple synthetic ground truth parameters and perform inference on the simulated datasets. For each dataset, we compute the rank of the ground truth within the posterior samples and summarize the results using empirical cumulative distribution function (ECDF) difference plots – a procedure known as simulation-based calibration (SBC) checking (Talts et al., 2020; Säilynoja et al., 2022; Modrák et al., 2023). SBC assesses whether the inference method is calibrated by verifying that the rank of the true parameter among posterior samples follows a uniform distribution across repeated simulations. To compare runtimes, we estimated how many observation sets are needed before the accumulated computational cost of repeated E-Post inference runs exceeds the total cost of UA-SABI, including both training and its repeated (quasi-instant) inference.

**General Setup**  For all case studies, we use polynomial chaos expansion (PCE) for the surrogate model (Wiener, 1938; Sudret, 2008; Oladyshkin & Nowak, 2012), as it is a fast-to-evaluate and fast-to-train surrogate and a flexible black-box model which can be applied to (almost) arbitrary simulation models. In principle, PCE is just one example among many possible choices – any uncertainty-aware surrogate could be used and replaced in a modular fashion depending on the application domain. A deterministic PCE constructs the surrogate through a spectral projection onto orthogonal (w.r.t. $p(\mathbf{x}, \boldsymbol{\omega})$) polynomial basis functions, expressed as

$$\widetilde{M_\mathbf{c}}(\mathbf{x}, \boldsymbol{\omega}) = \sum_{j=0}^{J} c_j \cdot \Psi_j(\mathbf{x}, \boldsymbol{\omega}), \tag{14}$$

with $\boldsymbol{\Psi} = \{\Psi_j\}_{j=0}^{J}$ being the multivariate orthogonal polynomial basis and $\mathbf{c} = \{c_j\}_{j=0}^{J}$ the corresponding coefficients. The number of expansion terms $J$ is computed via the standard truncation scheme (Sudret, 2008). For our case studies, we choose a relatively low polynomial degree, as we operate in a data-scarce regime and aim to avoid overfitting. In the real-world case studies, the underlying models are highly computationally expensive, so a higher polynomial degree would require additional model runs that could exceed the available computational budget. In general, selecting the appropriate surrogate complexity is important but challenging to determine a priori, as it depends on the specific application.

To construct a Bayesian PCE (Shao et al., 2017; Bürkner et al., 2023), we define a prior distribution over the surrogate coefficients $p(\mathbf{c})$ and the approximation error parameter $p(\sigma)$. We choose a normal likelihood for the simulator output $\mathbf{y} = M(\mathbf{x}, \boldsymbol{\omega})$:

$$p(\mathbf{y} \mid \mathbf{x}, \boldsymbol{\omega}, \mathbf{c}, \sigma) = \mathcal{N}(\mathbf{y} \mid \widetilde{M_\mathbf{c}}(\mathbf{x}, \boldsymbol{\omega}), \sigma). \tag{15}$$

Employing Hamiltonian Monte Carlo (Hoffman & Gelman, 2014) yields samples from the surrogate posterior $p(\mathbf{c}, \sigma \mid D_T)$ (Bürkner et al., 2023). The resulting Bayesian PCEs serve as our uncertainty-aware surrogate.

## 4.2 Case Study 1: LogSin Model

We first evaluate our surrogate-based ABI approaches in a simple synthetic scenario: a simulation model with one parameter $\omega$ and a one-dimensional input $x$:

$$y = M(\mathrm{x}, \omega) = \omega \log(\mathrm{x}) + \sin(0.05\mathrm{x}) + 0.01\mathrm{x} + 1. \tag{16}$$

### 4.2.1   Setup

We generate surrogate training data by evaluating the simulation model $M$ at the first 16 points of a two-dimensional Sobol sequence (Sobol', 1967), scaled to $[1, 200]$ for x and $[0.6, 1.4]$ for $\omega$. The resulting input-output pairs are used to train a Bayesian PCE. This introduces significant surrogate uncertainty and approximation error, as a perfect match to the simulation model is unattainable. The surrogate's posterior is sampled using HMC via *Stan* (Carpenter et al., 2017; Stan Development Team, 2024). We employ 4 chains, each with 1,000 warm-up and 250 sampling iterations, yielding 1,000 surrogate posterior samples in total to be propagated.

To perform inference on observation sets, we sample 200 ground truth parameters $\omega^*$ from a prior $p(\omega) = \mathcal{N}(1, 0.2)$ and generate four $(\mathbf{x}, \mathbf{y})$ observations for each. We compare the inference results of our proposed surrogate-based ABI to surrogate-based MCMC methods. For ABI, we used an equivariant *DeepSet* summary network (Zaheer et al., 2017), given $p(\mathbf{y} \mid \mathbf{x})$ being i.i.d., and a coupling flow inference network. We employ online training, where newly sampled surrogate outputs are used at each iteration. All ABI models were trained for 100 epochs, with a batch size of 64 and 128 batches per epoch, using *BayesFlow* (Radev et al., 2023b). For given observations, ABI generates 4,000 posterior samples via the inference network. Surrogate-based MCMC (Point) draws 4,000 samples using 4 chains (1,000 warm-up and 1,000 sampling iterations). For E-Post, we run MCMC separately for each surrogate posterior draw (1,000 warm-up, 4 sampling iterations per run), a process that is embarrassingly parallelizable but still computationally expensive. Further computational details are given in Appendix C.

### 4.2.2   Impact of Uncertainty Propagation

First, we compare the performance of our two surrogate-based methods, SABI and UA-SABI, against each other. Then we compare them against a low-budget ABI, trained with the same simulation data as used for surrogate training, and a full-budget ABI trained on sufficient simulation data. Figure 4a shows corresponding recovery plots. They show the posterior median (circles) and median deviation (vertical lines) for any given ground truth value $\omega^*$.

Figure 4a shows that a low-budget ABI model is unable to recover the ground truth values, particularly towards the boundaries of the parameter domain. Using a surrogate as part of SABI or UA-SABI, it becomes possible to generate sufficient ABI training data. Assigning the inter- and extrapolation tasks to the surrogate, specifically designed for this purpose, produces better outcomes than relying on the NPE to implicitly interpolate the posterior from scarce data. However, without propagating surrogate uncertainty (i.e., using SABI), estimated posterior uncertainty remains minimal and does not cover the true values, thus producing overconfident posteriors. Conversely, incorporating surrogate uncertainty via UA-SABI leads to an increase in posterior uncertainty, now better covering the true values. Indeed, the ECDF difference plots showing calibration in Fig. 4b confirm that both low-budget ABI and SABI produce severely miscalibrated posteriors, while UA-SABI and full-budget ABI yield well-calibrated posteriors. However, achieving this calibration with full-budget ABI requires substantially more training data, highlighting the efficiency of UA-SABI under limited simulation budgets.

### 4.2.3   Validation of Parameter Posterior

We validate the correctness of the surrogate-based ABI approaches and benchmark their performance against a reference solution. To this end, we compare the two surrogate-based ABI methods, SABI and UA-SABI, against the corresponding surrogate-based MCMC methods, Point and E-Post. We plot the posterior medians (circles) and their median deviations (lines) for the ABI methods along with the corresponding MCMC results, as well as the ECDF differences for all the methods in Fig. 5.

When comparing the ABI posteriors to the MCMC full reference solution, we observe that for the uncertainty-unaware methods, the SABI results align closely with the MCMC counterpart. Comparing the respective ECDF differences it shows that both Point and SABI produce similarly overconfident posteriors. For the uncertainty-aware methods, E-Post and UA-SABI, we observe a slight difference in the recovery plots and the ECDF differences. To investigate this discrepancy, we performed a convergence analysis for the MCMC

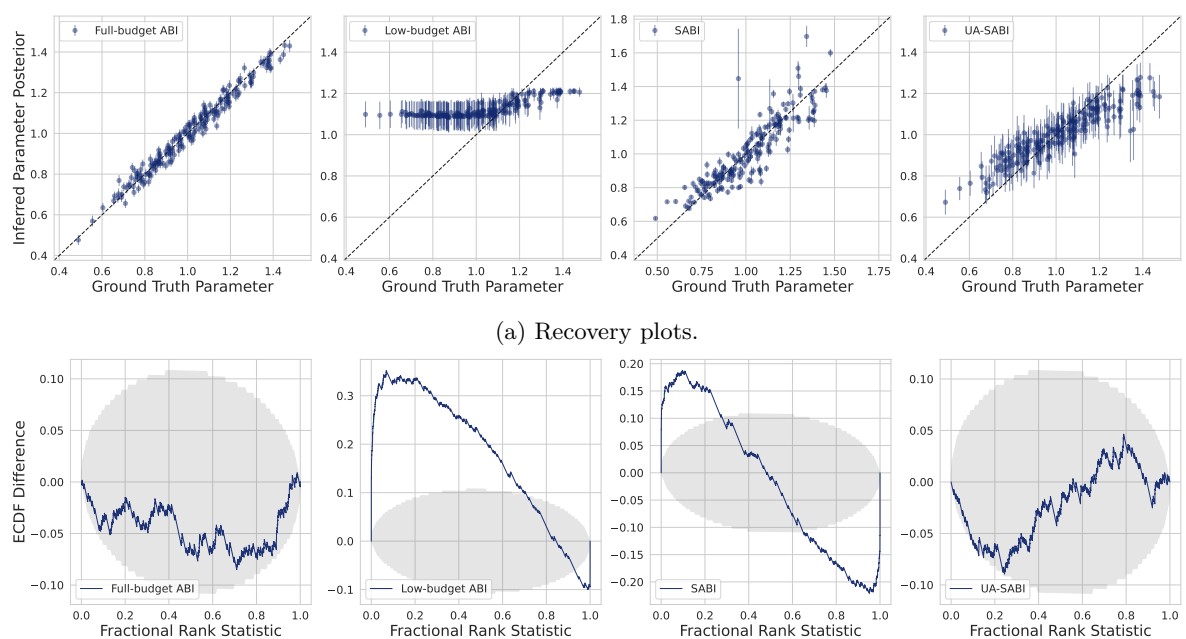

(a) Recovery plots.

(b) ECDF difference plots. Only full-budget ABI and UA-SABI are well calibrated.

Figure 4: LogSin recovery plots (top) and ECDF difference plots (bottom) for full-budget ABI, low-budget ABI, SABI, and UA-SABI (from left to right) over 200 ground truth samples. In the ECDF difference plots, empirical ranks are shown in blue, 95% confidence bands assuming calibration are shown in grey.

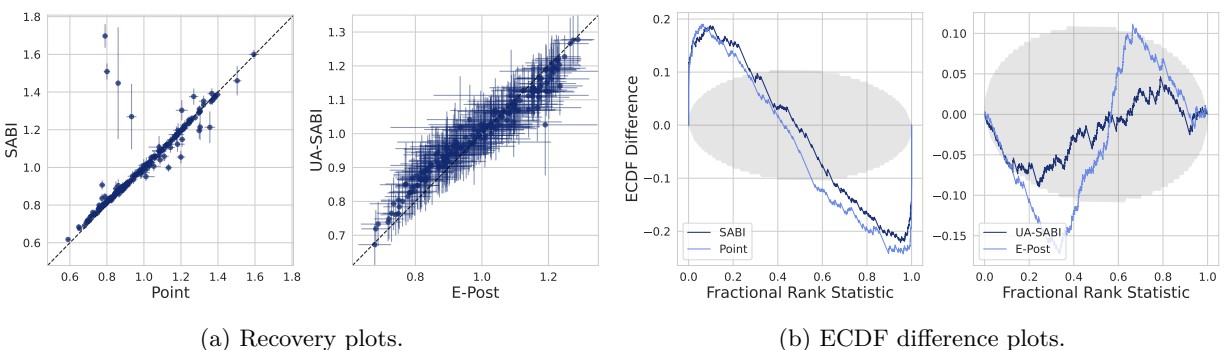

(a) Recovery plots.

(b) ECDF difference plots.

Figure 5: LogSin recovery plots (left) and ECDF difference plots (right) comparing ABI methods to corresponding MCMC methods over 200 ground truth samples. In the ECDF difference plots, empirical ranks are shown in blue, 95% confidence bands assuming calibration are shown in grey.

sampling of E-Post, with results shown in Appendix D. The analysis confirms that the difference can be attributed to non-convergence of many MCMC runs. Examining the calibration results reveals that while E-Post is slightly underconfident, UA-SABI yields well-calibrated posteriors. This indicates the reliability and correctness of UA-SABI.

### 4.2.4 Runtime Comparison

Next, we justify the computational effort required to train UA-SABI. Specifically, we aim to determine the break-even point – that is, the number of observation sets (i.e., measurement series) after which training UA-SABI and performing (quasi-instant) inference becomes more efficient than repeatedly rerunning E-Post. Beyond that point, the one-time training cost of UA-SABI is amortized. All experiments were performed

on a standard laptop equipped with an Intel Core i7-1185G7 CPU. For ABI, we consider both the upfront training phase and the inference time for the given sets as part of the total runtime. For E-Post, we measure the inference runtime while parallelized across 8 cores. Unlike for UA-SABI, E-Post runtimes depend on the number of cores used for parallelization.

In Fig. 7a, we show the runtime of UA-SABI and E-Post for $\{5, 10, 15, 20\}$ inference runs. We observe that UA-SABI's runtime is nearly constant for the number of inference runs, while E-Post scales linearly. Based on this comparison, UA-SABI is already justified after around 9 inference runs.

### 4.3   Case Study 2: Carbon Dioxide ($CO_2$) Storage Model

In the second case study, we test our method on a virtual benchmark that represents a real-world problem. Specifically, we consider a $CO_2$ storage benchmark (Köppel et al., 2019). In this test case, a non-linear hyperbolic partial differential equation models the two-phase flow of $CO_2$ in brine. It describes $CO_2$ injection, plume migration, pressure build-up, and the influence of uncertain porous medium properties in a deep saline aquifer. Given the $CO_2$ saturation as measurements, our aim is to infer three parameters of interest: injection rate of $CO_2$ (IR), relative permeability degree in the fractional flux function (PM), and porosity (PR) of the formation. This test case allows us to analyze the performance of UA-SABI in inference for hydrosystem models. The underlying physical model exhibits strong discontinuities at certain combinations of parameter values and space/time coordinates, which inherently limit the approximation performance of PCE-based surrogate models. However, this scenario provides an opportunity to test UA-SABI under challenging conditions where the surrogate model is strongly misspecified. Five separate Bayesian PCE models with sparsity-inducing priors (Bürkner et al., 2023) are trained for five different instants at days $\{20, 40, 60, 80, 100\}$ and at a fixed location 30m from the injection well. A similar setup was studied in (Oladyshkin et al., 2020). We show parameter posterior results for porosity, the most informative parameter of the $CO_2$ storage model.

#### 4.3.1   Setup

In general, the setup for the $CO_2$ storage model follows the same structure as that described in Section 4.2.1. The priors of the input parameters are given as IR $\sim 6.4 \times 10^{-4} \times (1 + \text{Beta}(4, 2))$, PM $\sim 2 \times \text{Beta}(1.25, 1.25) + 2$, and PR $\sim \text{Beta}(2.4, 9)$ (Köppel et al., 2019). For PCE training, we considered 64 Sobol sequence evaluations scaled to the input parameter prior ranges and constructed the polynomial basis with arbitrary polynomial chaos (aPC) based on the priors (Oladyshkin & Nowak, 2012; Bürkner et al., 2023).

In contrast to case study 1, we used an offline ABI training set, where the training data is pre-generated from $10^4$ parameter prior draws and fixed during the training process, allowing for direct comparison with a standard full-budget ABI trained on pre-generated simulation data. Further computational details are given in Appendix C.

#### 4.3.2   Validation of Parameter Posterior

We compare the inferred posterior samples obtained from a full-budget ABI ($N_B = 10^4$), a low-budget ABI ($N_B = 64$), SABI, and UA-SABI. Full-budget ABI was feasible due to the already available data (Köppel et al., 2017; Oladyshkin et al., 2020). Therefore, recovery plots were chosen to validate the quality of our results relative to standard ABI (Fig. 6).

Recovery plots in Fig. 6a again show that low-budget ABI struggles to recover the ground truth parameter, particularly near domain boundaries. SABI also produces poor estimates and, moreover, fails to capture uncertainty in the inferred parameter posteriors. This overconfidence results in miscalibrated posteriors, confirmed by the two corresponding ECDF difference plots in Fig. 6b.

In contrast, UA-SABI performs comparably to full-budget ABI while accounting for the additional uncertainty introduced by using a surrogate to generate training data. According to the ECDF difference plots, it produces a well-calibrated posterior for porosity. Additionally, recovery plots between ABI and MCMC for surrogate-based methods and ECDF difference plots for MCMC-based methods are provided in Appendix E.

Despite the model's discontinuities and the resulting surrogate misspecification, our method successfully quantifies and propagates surrogate uncertainty to the parameter posterior.

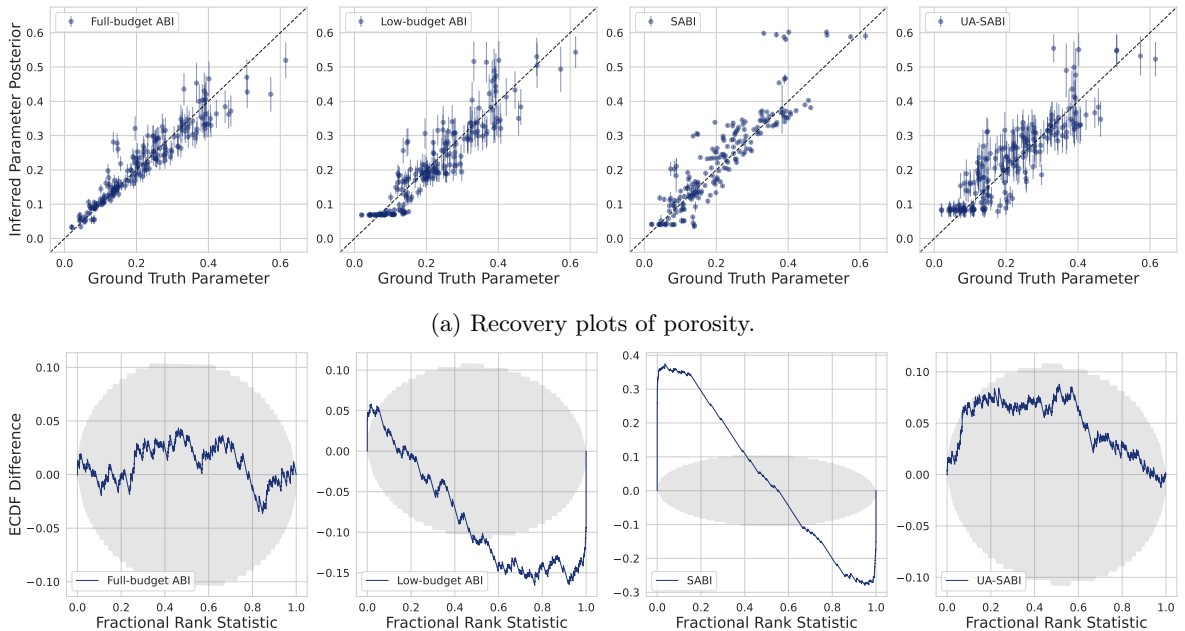

(a) Recovery plots of porosity.

(b) ECDF difference plots of porosity. Only full-budget ABI and UA-SABI are well calibrated.

Figure 6: $CO_2$ recovery plots (top) and ECDF difference plots (bottom) for full-budget ABI, low-budget ABI, SABI, and UA-SABI (from left to right) over 200 ground truth samples. In the ECDF difference plots, empirical ranks are shown in blue, 95% confidence bands assuming calibration are shown in grey.

### 4.3.3   Runtime Comparison

Also, for the $CO_2$ storage model, we compare the runtimes, following the same approach as in Section 4.2.4. The experiments were run on a computing cluster with two AMD EPYC 7551 CPUs (totaling 64 physical cores) to speed up E-Post through parallelization.

Figure 7b presents the measured runtimes of UA-SABI and E-Post for $\{5, 6, 7, 8\}$ inference runs, whereby E-Post was parallelized on 16 cores. We observe a break-even point between 6 and 7 inference runs, indicating that UA-SABI becomes justified after 7 runs. Despite using more cores for E-Post, the training costs are amortized earlier for a more expensive model.

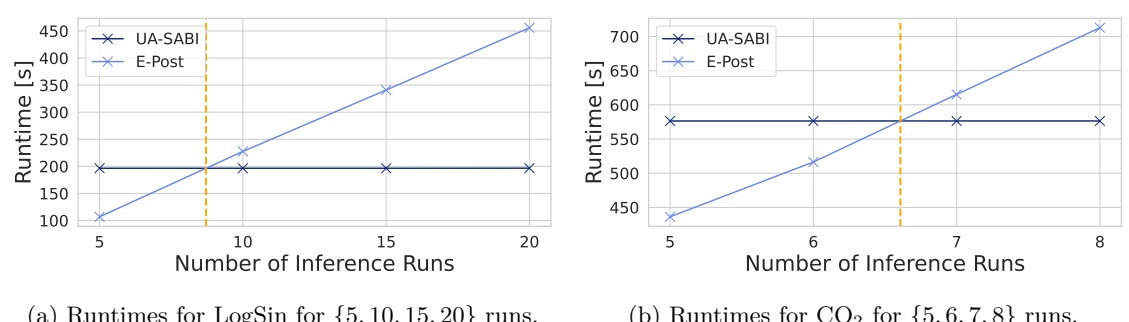

(a) Runtimes for LogSin for $\{5, 10, 15, 20\}$ runs.     (b) Runtimes for $CO_2$ for $\{5, 6, 7, 8\}$ runs.

Figure 7: Comparison of runtimes for UA-SABI (training and inference) and E-Post (inference) with break-even point.

### 4.4 Case Study 3: Microbially Induced Calcite Precipitation Model

In this case study, we transition from benchmarks to realistic conditions by using real measurement data. Case studies 1 and 2 introduced a toy example and a virtual benchmark, respectively. Case study 3 now focuses on posterior inference from observed measurements, where the primary goal is to evaluate the applicability of UA-SABI under real-world conditions. Specifically, in addition to simulation-to-surrogate misspecification, we encounter simulation-to-real and thus surrogate-to-real misspecification. A runtime comparison for the MICP model on synthetic data is added in Appendix F.5.

The underlying model predicts microbially induced calcite precipitation (MICP). MICP is a reactive transport process, here taking place inside a porous medium. It includes at least biofilm, calcite, and the unreactive solid matrix as solid phases, water, and, in some cases, an additional fluid phase such as gas. Relevant in the aqueous phase are dissolved calcium, inorganic carbon, and urea. The complete set of relevant species is detailed in Hommel et al. (2015). The process is mediated by S. pasteurii, a bacterium that produces the enzyme urease. Urease catalyzes the hydrolysis of urea into ammonia and carbonic acid. In aqueous solution, ammonia consumes hydrogen ions ($H^+$), which raises the pH. The increase in pH alters carbonate equilibria: carbonic acid dissociates further, releasing additional $H^+$ and carbonate ions. This raises the concentration of dissolved carbonate species. When calcium ions are present, they combine with carbonate ions and trigger the precipitation of calcite within the pore space. A schematic illustration of the relevant process during MICP is given in Appendix F.1.

The experiment is performed on a 61cm sand-filled column with a 2.54cm diameter. Bacteria are first injected from the bottom, followed by an overnight no-flow period to allow biofilm formation. Biofilm growth is then stimulated with a 24h substrate injection. Afterwards, two pore volumes of a 0.33mol/L calcium-urea solution are injected at 10ml/min, repeated every 24h. Each injection is followed by a no-flow period for mineralization, and then a substrate injection to reactivate the biofilm (Hommel et al., 2015). This cycle is repeated 30 times. A more detailed description can be found in Hommel et al. (2015).

Measurements of the volume fraction of calcite on eight spatial points located at 3.81, 11.43, 19.05, 26.67, 34.29, 41.91, 49.53, and 57.15cm distance from the bottom are available. After the last cycle, we build our surrogate based on four parameters of interest: the coefficient for preferential attachment to biomass ($c_{a,1}, [\mathrm{s}^{-1}]$), the coefficient for attachment to arbitrary surfaces ($c_{a,2}, [\mathrm{s}^{-1}]$), the dry mass density of biofilm ($\rho_f, [\mathrm{kg/m}^3]$), and the enzyme content of biomass ($k_{ub}, [\mathrm{kg/kg}]$).

The MICP case provides an opportunity to test UA-SABI on a model that is highly computationally expensive yet low-dimensional in parameter space, and for which real-world measurement data are available. We again employ PCE as the surrogate modeling technique, due to its demonstrated success for this model (Scheurer et al., 2021). In total, eight Bayesian PCEs are trained, one for each measurement location.

#### 4.4.1 Setup

In general, the setup for the MICP model follows the same structure as that described in the former case studies. The priors of the input parameters are given as $c_{a,1} \sim \mathcal{U}(10^{-10}, 10^{-7})$, $c_{a,2} \sim \mathcal{U}(10^{-10}, 10^{-6})$, $\rho_f \sim \mathcal{U}(1, 15)$, and $k_{ub} \sim \mathcal{U}(10^{-5}, 5 \cdot 10^{-4})$ (Scheurer et al., 2021). For PCE training, 25 model evaluations within the prior ranges of input parameters are available, which are not necessarily optimal for surrogate training. We constructed the polynomial basis with aPC based on the available model evaluations (Oladyshkin & Nowak, 2012; Bürkner et al., 2023).

As in case study 2, we used an offline ABI training set of $10^4$ parameter prior draws. In contrast to case study 2, no pre-generated simulation data are available for these draws. Instead, output data can only be generated through the surrogate; thus, no comparison with a full-budget ABI is possible. Further computational details are given in Appendix C.

#### 4.4.2 Validation of Surrogate Model and Parameter Inference Setup

Since inference will be performed using real measurement data, the first step is to evaluate how accurately the surrogate represents the underlying system. Figure 8a presents the measurement data, the model evaluations

used for surrogate training, and the surrogate predictions obtained from $10^4$ prior samples across the eight locations. The results indicate that the simulated model data—and consequently the surrogate – do not cover the region occupied by the measurement data. This makes the measurement data out-of-distribution (OOD) relative to the simulation model's a priori predictions – a phenomenon that is suspected to occur frequently in dynamic systems modeling, although rarely investigated in detail.

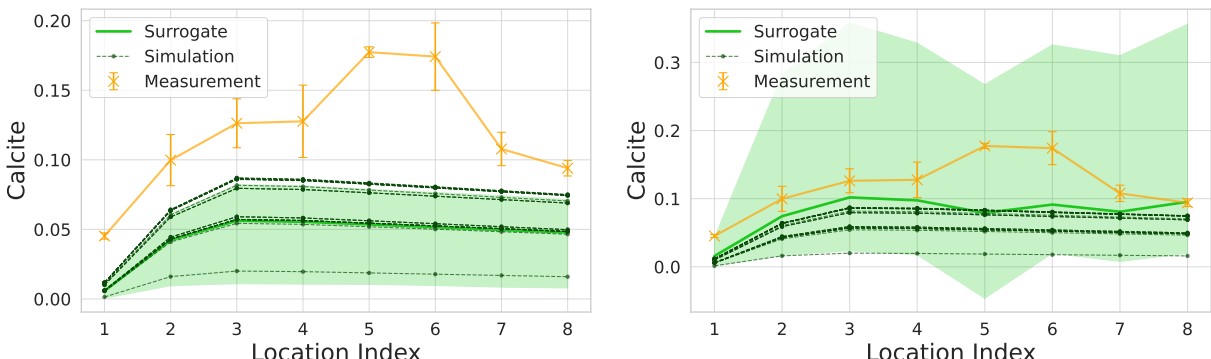

(a) Surrogate output with original prior for $k_{ub} \sim \mathcal{U}(10^{-5}, 5 \cdot 10^{-4})$.

(b) Surrogate output with adjusted prior for $k_{ub} \sim \mathcal{U}(10^{-5}, 5 \cdot 10^{-3})$.

Figure 8: Measurement data for calcite, 25 model evaluations, and surrogate model output obtained with original prior (left) and adjusted prior (right) for $k_{ub}$ of the MICP model across eight locations. Surrogate predictions are obtained from $10^4$ prior samples; median and 90% confidence intervals are shown.

Neural networks are known to generalize poorly outside their training domain, resulting in poor posteriors obtained via ABI (Schmitt et al., 2024a), which is shown in Appendix F.2. Thus, we seek to transfer this challenge to the surrogate instead. To this end, we found that the surrogate priors used for parameter inference can be adjusted to ensure that the measurement data fall within the surrogate's support. This approach leverages the generalization properties of the low-dimensional, regularized surrogate, avoiding dependence on the poor generalization abilities of neural networks, which are used within the ABI model during inference.

The results of a sensitivity analysis (more detailed in Appendix F.3) on the surrogate using Sobol indices (Sobol', 1990) shows that the surrogate's output at all locations is almost only sensitive to $k_{ub}$.

In Fig. 9, the surrogate output is shown exemplarily at location index 3 by varying $k_{ub}$. It can be observed that the uncertainty of the surrogate increases when evaluating OOD inputs. Based on these results, we adapt the parameter inference prior range for $k_{ub}$, such that $k_{ub} \sim \mathcal{U}(10^{-5}, 5 \cdot 10^{-3})$. This adjustment results in a prior predictive that covers the measurement data, as illustrated in Fig. 8b.

### 4.4.3 Comparison of Parameter Posterior

Since parameter inference is carried out on real measurement data, no ground truth is available. We therefore compare only the inferred posteriors obtained using low-budget ABI ($N_B = 25$), SABI, UA-SABI, Point, and E-Post. For easy visual comparison, we applied kernel density estimation (KDE) with Scott's bandwidth method to the posterior samples from each approach. Since $k_{ub}$ is the only sensitive parameter, we present its posterior estimates in Fig. 10 to avoid redundancy, while posterior estimates for the remaining parameters are provided in Appendix F.4.

Figure 10 demonstrates that low-budget ABI, which is shown as a point estimate due to quasi-nonexistent variance in the posterior, exhibits a pronounced bias and substantially underestimates posterior uncertainty. This confirms again that training an ABI model with very limited data is not meaningful. Comparing the surrogate-based methods further highlights that explicitly quantifying surrogate uncertainty and propagating it through the inference process is reflected in the wider posterior estimates. However, E-Post and UA-SABI posteriors exhibit slight difference, which we attribute to non-convergence of some MCMC runs in E-Post.

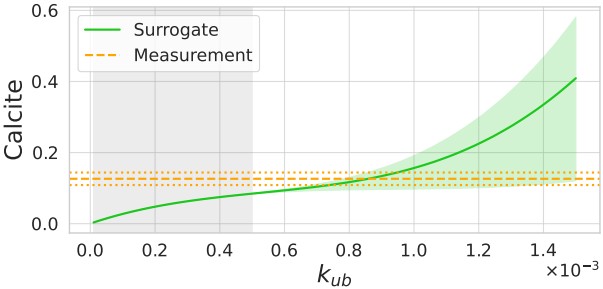

Figure 9: MICP surrogate output at location index 3 over $k_{ub}$. The output is obtained by evaluating the uncertainty-aware surrogate for $k_{ub} \in [10^{-5}, 5 \cdot 10^{-3}]$ and keeping the other parameters fixed at their means. Median and 90% confidence intervals of the surrogate output are shown in green, the region of the original prior is shown in grey.

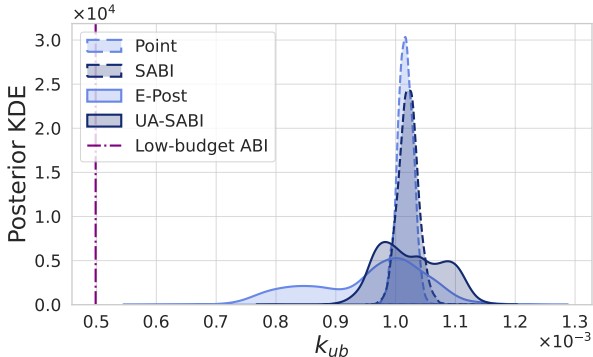

Figure 10: KDEs of 4,000 parameter posterior samples for $k_{ub}$ of the MICP model obtained with low-budget ABI, Point, E-Post, SABI, and UA-SABI.

To assess the validity of the obtained posteriors and the appropriateness of the quantified and propagated uncertainty, Fig. 11 presents the posterior predictive distributions for calcite across the four surrogate-based methods.

Figure 11a and Fig. 11b clearly illustrate that not quantifying and propagating uncertainty results in over-confident posterior predictives. In contrast, as shown in Fig. 11c and Fig. 11d, explicitly accounting for uncertainty produces confidence intervals that successfully cover the measurement data. Both methods yield comparable intervals, with UA-SABI producing a slightly broader interval as its higher posterior density in the out-of-distribution region of the most sensitive parameter $k_{ub}$ (Fig. 10) yields increased surrogate predictive uncertainty.

Overall, this demonstrates that quantifying and propagating the uncertainty of a surrogate model enables ABI for highly computationally expensive models. Even when the available model simulations are suboptimal for surrogate training and far from the measurement data, accounting for uncertainty and promoting generalization in the surrogate model leads to more trustworthy results.

## 5   Summary and Outlook

In this work, we introduced Uncertainty-Aware Surrogate-based Amortized Bayesian Inference (UA-SABI) – a framework designed to enable efficient and reliable ABI for computationally expensive models. UA-SABI combines surrogate modeling and ABI while explicitly quantifying and propagating surrogate uncertainties through the inference process. This addresses a core limitation of existing approaches: while surrogate

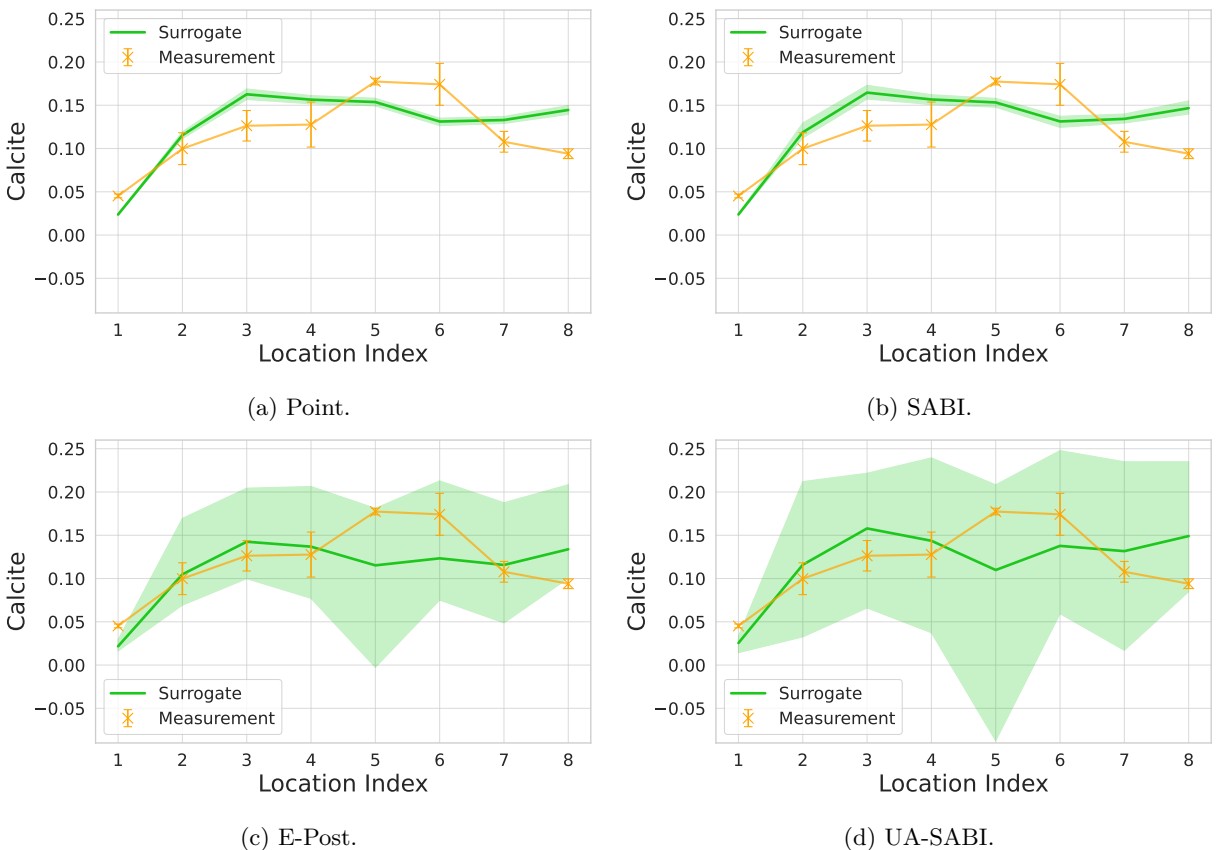

Figure 11: MICP posterior predictives for calcite for the four surrogate-based methods. Surrogate predictions are obtained from 4,000 posterior samples, median and 90% confidence intervals are shown.

models can reduce the cost of generating training data for ABI, ignoring their approximation error often leads to overconfident and misleading posteriors. By incorporating uncertainty awareness, UA-SABI enables better-calibrated and reliable inference, even under tight computational constraints.

We validated UA-SABI in both a simple toy example and two real-world problems of modeling $CO_2$ storage and MICP, highlighting its ability to produce well-calibrated posterior estimates that match those obtained with MCMC-based methods. Our experiments demonstrate the importance of sufficient training data, as low-budget ABI produces erroneous posteriors. Also, they show the importance of uncertainty propagation when using a surrogate: Surrogate-based ABI results in overconfident posteriors, whereas UA-SABI correctly reflects model uncertainty in its predictions. Moreover, we showed that the upfront computational effort of training UA-SABI is quickly offset in scenarios involving repeated inference, with amortization becoming beneficial already after a few inference runs. Additionally, the MICP experiment poses a challenging real-world scenario by having 1) only a limited number of model runs available, and 2) real measurement data that are not adequately captured by these model runs.

Overall, UA-SABI offers an efficient and reliable solution in settings that require repeated inference for computationally expensive models. In general, we conclude that using surrogates to generate training data can be an effective strategy for ABI when simulations are scarce or computationally expensive. Moreover, explicitly accounting for surrogate uncertainty in the training data improves ABI results, particularly by reducing overconfidence.

In this work, we applied UA-SABI only to low-dimensional, computationally expensive problems to keep results interpretable and avoid challenges that would distract from our scope. While this focus allowed us to clearly demonstrate the method's core advantages, the question of scalability to higher dimensions naturally

arises. In general, standard ABI performs well in high-dimensional settings when sufficient simulated training data is available (e.g. Zhou et al., 2025). However, data-scarcity necessitates the use of surrogate models to augment the limited training data. Extending UA-SABI to high-dimensional problems thus hinges on developing surrogates that satisfy two critical requirements: robustness to high-dimensionality and minimal training data requirements. If these conditions cannot be met, standard ABI remains the more practical choice. High-dimensional surrogate modeling, however, presents fundamental challenges. In the PCE framework, for example, a high-dimensional parameter space results in a high-dimensional coefficient space and thus a high-dimensional integration problem, which suffers from the curse of dimensionality. Consequently, the surrogate type must be chosen to accommodate this challenge. Neural network surrogates could be one option, but they require substantial training data again (similar to standard ABI) so nothing is gained here. In PCE, existing approaches try to address high-dimensionality through input dimensionality reduction techniques (e.g. Li & Tartakovsky, 2020) or sparsity methods in PCE (e.g. Bürkner et al., 2023; Luthen et al., 2021) to reduce either the parameter space or the integration space. Combining these methods would be a valuable direction for future research.

Additionally, the surrogate is technically misspecified, as the simulator is not fully contained within the surrogate class. Thus, future work could integrate UA-SABI with recently proposed methods for detecting and mitigating model misspecification in ABI (Schmitt et al., 2024a; Dellaporta et al., 2022; Elsemüller et al., 2025), which are all naturally compatible with our framework.

## Author Contributions

This work represents an equal contribution of Stefania Scheurer and Philipp Reiser. Following the CRediT taxonomy:
**Stefania Scheurer:** Conceptualization, Methodology, Software, Writing - Original Draft, Writing - Review & Editing, **Philipp Reiser:** Conceptualization, Methodology, Software, Writing - Original Draft, Writing - Review & Editing, **Tim Brünnette:** Writing - Review & Editing, **Wolfgang Nowak:** Writing - Review & Editing, **Anneli Guthke:** Writing - Review & Editing, **Paul-Christian Bürkner:** Conceptualization, Writing - Review & Editing.

## Acknowledgments

We thank the Deutsche Forschungsgemeinschaft (DFG, German Research Foundation) for supporting this work by funding – EXC2075 – 390740016 under Germany's Excellence Strategy and the Collaborative Research Centre SFB 1313, Project Number 327154368. We acknowledge the support by the Stuttgart Center for Simulation Science (SimTech). We further acknowledge the support of the DFG Collaborative Research Center 391 (Spatio-Temporal Statistics for the Transition of Energy and Transport) – 520388526. Additionally, we thank Dr. Ilja Kröker for his help and to both Dr. Kröker and Dr. Johannes Hommel for providing data.

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

# A Algorithmic Overview

Algorithm 1 gives an algorithmic overview of UA-SABI training. Note that this is for online training. In case of offline training, inputs and parameters $\{(\mathbf{x}^{(i)}, \boldsymbol{\omega}^{(i)})\}_{i=1}^{N_B}$ are fixed and not resampled in each epoch.

---

**Algorithm 1** Training of Surrogate and UA-SABI

---

    **Surrogate Training Phase**
**Require:** Simulation model $M(\mathbf{x}, \boldsymbol{\omega})$, prior $p(\mathbf{c}, \sigma)$, likelihood $p(\mathbf{y} \mid \mathbf{x}, \boldsymbol{\omega}, \mathbf{c}, \sigma)$
**Ensure:** Posterior $p(\mathbf{c}, \sigma \mid D_T)$
  1: Choose inputs and parameters $\{(\mathbf{x}^{(i)}, \boldsymbol{\omega}^{(i)})\}_{i=1}^{N_T}$
  2: **for** $i = 1, \ldots, N_T$ **do**
  3:     Evaluate simulation model $\mathbf{y}^{(i)} = M(\mathbf{x}^{(i)}, \boldsymbol{\omega}^{(i)})$
  4: **end for**
  5: Construct training dataset $D_T = \{(\mathbf{x}^{(i)}, \boldsymbol{\omega}^{(i)}, \mathbf{y}^{(i)})\}_{i=1}^{N_T}$
  6: Perform Bayesian inference for surrogate parameters

$$p(\mathbf{c}, \sigma \mid D_T) \propto \prod_{i=1}^{N_T} p(\mathbf{y}^{(i)} \mid \mathbf{x}^{(i)}, \boldsymbol{\omega}^{(i)}, \mathbf{c}, \sigma) \, p(\mathbf{c}, \sigma)$$

    **UA-SABI Training Phase**
**Require:** Prior $p(\mathbf{x}, \boldsymbol{\omega})$, posterior $p(\mathbf{c}, \sigma \mid D_T)$
**Ensure:** Trained summary and inference network parameters $\widehat{\boldsymbol{\theta}}, \widehat{\boldsymbol{\varphi}}$
  7: **for** each epoch **do**
  8:     **for** $i = 1, \ldots, N_B$ **do**
  9:         Sample inputs and parameters from prior $(\mathbf{x}^{(i)}, \boldsymbol{\omega}^{(i)}) \sim p(\mathbf{x}, \boldsymbol{\omega})$
10:         Sample surrogate parameters from posterior $(\mathbf{c}^{(i)}, \sigma^{(i)}) \sim p(\mathbf{c}, \sigma \mid D_T)$
11:         Evaluate surrogate $\widetilde{\mathbf{y}}^{(i)} = \widetilde{M}_{\mathbf{c}^{(i)}}(\mathbf{x}^{(i)}, \boldsymbol{\omega}^{(i)})$
12:         Sample corrected surrogate output $\widetilde{\mathbf{y}}_\epsilon^{(i)} \sim p(\widetilde{\mathbf{y}}_\epsilon \mid \widetilde{\mathbf{y}}^{(i)}, \sigma^{(i)})$
13:         Pass $(\mathbf{x}^{(i)}, \widetilde{\mathbf{y}}_\epsilon^{(i)})$ through summary network $\mathbf{s}^{(i)} = S_{\boldsymbol{\theta}}(\mathbf{x}^{(i)}, \widetilde{\mathbf{y}}_\epsilon^{(i)})$
14:         Pass $\mathbf{s}^{(i)}$ through inference network $I_{\boldsymbol{\varphi}}(\mathbf{s}^{(i)})$ which implies $q_{\boldsymbol{\varphi}}(\boldsymbol{\omega}^{(i)} \mid \mathbf{s}^{(i)})$
15:     **end for**
16:     Compute loss from Eq. (5) and update network parameters $\boldsymbol{\theta}, \boldsymbol{\varphi}$
17: **end for**

---

# B Proofs

We refer to ABI standard conditions as those under which ABI yields asymptotically correct posteriors, following Radev et al. (2020). Specifically, we assume (i) an infinitely large training dataset, (ii) a conditional neural density estimator $q_{\boldsymbol{\varphi}}(\boldsymbol{\omega} \mid \mathbf{y})$ that is sufficiently expressive, (iii) convergence of the training procedure to the true conditional density, and (iv) inference data drawn from the same data-generating process as the training data.

Likewise, MCMC standard conditions refer to ergodicity, irreducibility, and aperiodicity of the Markov chain (e.g., Robert & Casella, 2004), ensuring convergence of the chain to the target posterior distribution as the number of samples tends to infinity.

**Proposition 1.** *Under ABI standard conditions, and assuming an infinite number of samples used to propagate from the surrogate posterior $p(\mathbf{c}, \sigma \mid D_T)$, the posterior distribution targeted by UA-SABI converges to the E-Post posterior.*

*Proof.* In UA-SABI, as per Eq. (11), we sample from the joint distribution

$$p(\boldsymbol{\omega}, \mathbf{y}, \mathbf{c}, \sigma \mid D_T) = p(\mathbf{y} \mid \boldsymbol{\omega}, \mathbf{c}, \sigma) \, p(\boldsymbol{\omega}) \, p(\mathbf{c}, \sigma \mid D_T) \tag{17}$$

to train the conditional neural density estimator $q_{\boldsymbol{\varphi}}(\boldsymbol{\omega} \mid \mathbf{y})$. We only condition on the data $\mathbf{y}$ and treat the surrogate parameters $(\mathbf{c}, \sigma)$ as nuisance parameters ignored by $q_{\boldsymbol{\varphi}}(\boldsymbol{\omega} \mid \mathbf{y})$. This means that we (implicitly) integrate over the distribution of $(\mathbf{c}, \sigma)$. Hence, UA-SABI targets the following posterior:

$$p_{\text{UA-SABI}}(\boldsymbol{\omega} \mid \mathbf{y}, D_T) \propto \iint p(\mathbf{y} \mid \boldsymbol{\omega}, \mathbf{c}, \sigma)\, p(\boldsymbol{\omega})\, p(\mathbf{c}, \sigma \mid D_T)\, \mathrm{d}\mathbf{c}\, \mathrm{d}\sigma. \tag{18}$$

On the other hand, E-Post targets the following posterior, as per Eq. (13):

$$p_{\text{E-Post}}(\boldsymbol{\omega} \mid \mathbf{y}, D_T) = \iint p(\boldsymbol{\omega} \mid \mathbf{y}, \mathbf{c}, \sigma)\, p(\mathbf{c}, \sigma \mid D_T)\, \mathrm{d}\mathbf{c}\, \mathrm{d}\sigma \tag{19}$$

$$\propto \iint p(\mathbf{y} \mid \boldsymbol{\omega}, \mathbf{c}, \sigma)\, p(\boldsymbol{\omega})\, p(\mathbf{c}, \sigma \mid D_T)\, \mathrm{d}\mathbf{c}\, \mathrm{d}\sigma, \tag{20}$$

which shows $p_{\text{UA-SABI}}(\boldsymbol{\omega} \mid \mathbf{y}, D_T) = p_{\text{E-Post}}(\boldsymbol{\omega} \mid \mathbf{y}, D_T)$. $\qquad\square$

**Corollary 1.** *For an infinite number of samples from the surrogate posterior $p(\mathbf{c}, \sigma \mid D_T)$ and under ABI and MCMC standard conditions, the empirical distributions of UA-SABI samples and MCMC samples from E-Post converge to each other.*

## C  Additional Case Study Details

### C.1  Surrogate Models

In case study 1, we train a Bayesian PCE with 2-dimensional Legendre polynomials and a maximum total degree of 3, resulting in $J = 10$ polynomials. We set a normal prior for the surrogate coefficients, $p(\mathbf{c}) = \mathcal{N}(0, 5)$, and a half-normal prior for the approximation error parameter, $p(\sigma) = \text{Half-}\mathcal{N}(0.5)$.

In case study 2 and 3, we consider a Bayesian PCE with 3- and 4-dimensional aPC polynomials for the $CO_2$ model (see Section 4.3) and a maximum total degree of 3, resulting in $J = 19$ and $J = 35$ polynomials respectively. Following Bürkner et al. (2023), we place a sparsity-inducing R2D2 prior on the surrogate coefficients $\mathbf{c}$ with $R^2 \sim \text{Beta}(0.5, 2)$. For the approximation error we set the prior as $p(\sigma) = \text{Half-}\mathcal{N}(0.1)$.

### C.2  Neural Posterior Estimation

For the first two case studies, we used the same NPE setup. The summary network $S_{\boldsymbol{\theta}}(\mathbf{x}, \mathbf{y})$ is a *DeepSet* Zaheer et al. (2017) composed of two hidden layers, each containing 10 neurons. It outputs a 10-dimensional summary vector. For the inference network $I_{\boldsymbol{\varphi}}(\mathbf{s})$, we employ a coupling flow as implemented in Radev et al. (2023b). For the third case study, we did not use a summary network as the measurement locations in the experiment were fixed. For the inference network $I_{\boldsymbol{\varphi}}(\mathbf{s})$, we used a spline coupling flow (Durkan et al., 2019) as implemented in Radev et al. (2023b). The training process employed a cosine learning rate scheduler with an initial learning rate of $5 \times 10^{-4}$ and a minimum learning rate fraction of $\alpha = 10^{-6}$. The scheduler operated over a total of 12,800 steps, corresponding to 128 batches per epoch over 100 epochs. All NPE models were trained using the Adam optimizer (Kingma & Ba, 2017) for 100 epochs.

## D  Additional Results for Case Study 1: LogSin Model

This section shows the MCMC convergence analysis of E-Post for the LogSin model to explain the differences between UA-SABI and E-Post. We follow the same setup as described in Section 4.2.1, but used more samples and chains to check convergence. We analyzed the convergence for each of the 200 ground truth parameters and for each surrogate posterior sample respectively. We run MCMC separately for each of the 1,000 surrogate posterior samples with 4 chains and 1,000 warm-up and 1,000 sampling iterations, respectively, resulting in $1{,}000 \times 4 \times 1{,}000$ draws per ground truth parameter used for analysis.

For each surrogate posterior sample we can check the convergence of the MCMC run using standard diagnostics $\widehat{R}$. As convergence threshold, we follow the standard recommendation $\widehat{R} < 1.01$ (Vehtari et al., 2021).

Figure 12 shows the ratio of non-converged runs ($\widehat{R} > 1.01$) started by each surrogate posterior sample (SPS) vs. total number of runs for each of the 200 ground truth parameters. It is evident that for most ground truth parameter values more than 10% of the runs have not converged, with some having more than 40% of the runs with an $\widehat{R} > 1.01$. Notably, for most ground truth parameter values, more than 10% of the runs have not converged, with some parameters even exceeding more than 40% for that ratio. This indicates that the E-Post posterior is not reliable, which results in deviating posteriors for E-Post and UA-SABI, as shown in Fig. 5.

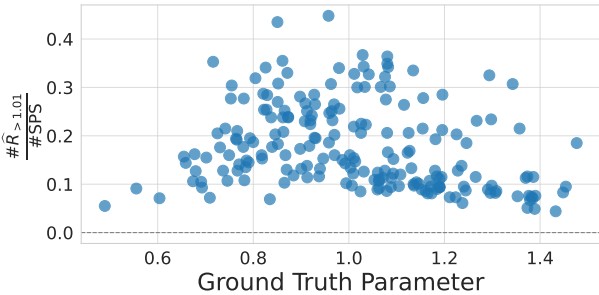

Figure 12: Ratio of non-converged runs ($\widehat{R} > 1.01$) started by each surrogate posterior sample (SPS) vs. total number of runs for each of the 200 ground truth parameters for the LogSin model.

Additionally, Fig. 13 shows the trace plots of single E-Post runs (4 chains) for two exemplary draws of the surrogate posterior with bad $\widehat{R}$ given a single ground truth parameter for the LogSin model. These traces indicate that the chains fail to mix properly between the two modes of the underlying posterior. When the modes have little overlap, MCMC struggles to recover their correct proportions.

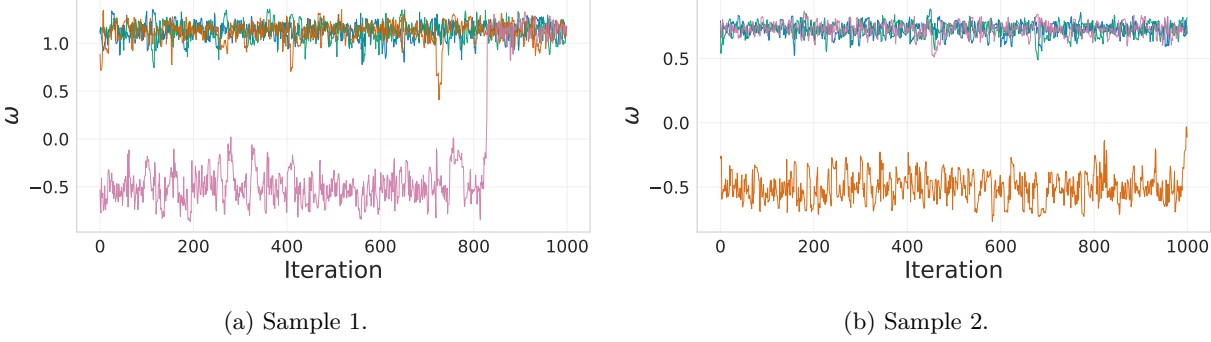

(a) Sample 1.          (b) Sample 2.

Figure 13: Trace plots of single E-Post MCMC runs for two exemplary draws of the surrogate posterior given a single ground truth parameter for the LogSin model. The different colors indicate different chains.

# E    Additional Results for Case Study 2: CO$_2$ Storage Model

This section shows the recovery and ECDF difference plots for porosity of the CO$_2$ storage model shown for SABI vs. Point and UA-SABI vs. E-Post.

In Fig. 14 we show that SABI and Point as well as UA-SABI and E-Post yield similar results in both recovery and calibration. The notably wide intervals observed for Point, especially compared to SABI in the recovery plots suggest convergence problems in the MCMC, likely due to the low standard deviation in the likelihood. These convergence problems may explain the inconsistencies between SABI and Point. In comparison, their uncertainty-aware counterparts UA-SABI and E-Post produce highly similar estimates.

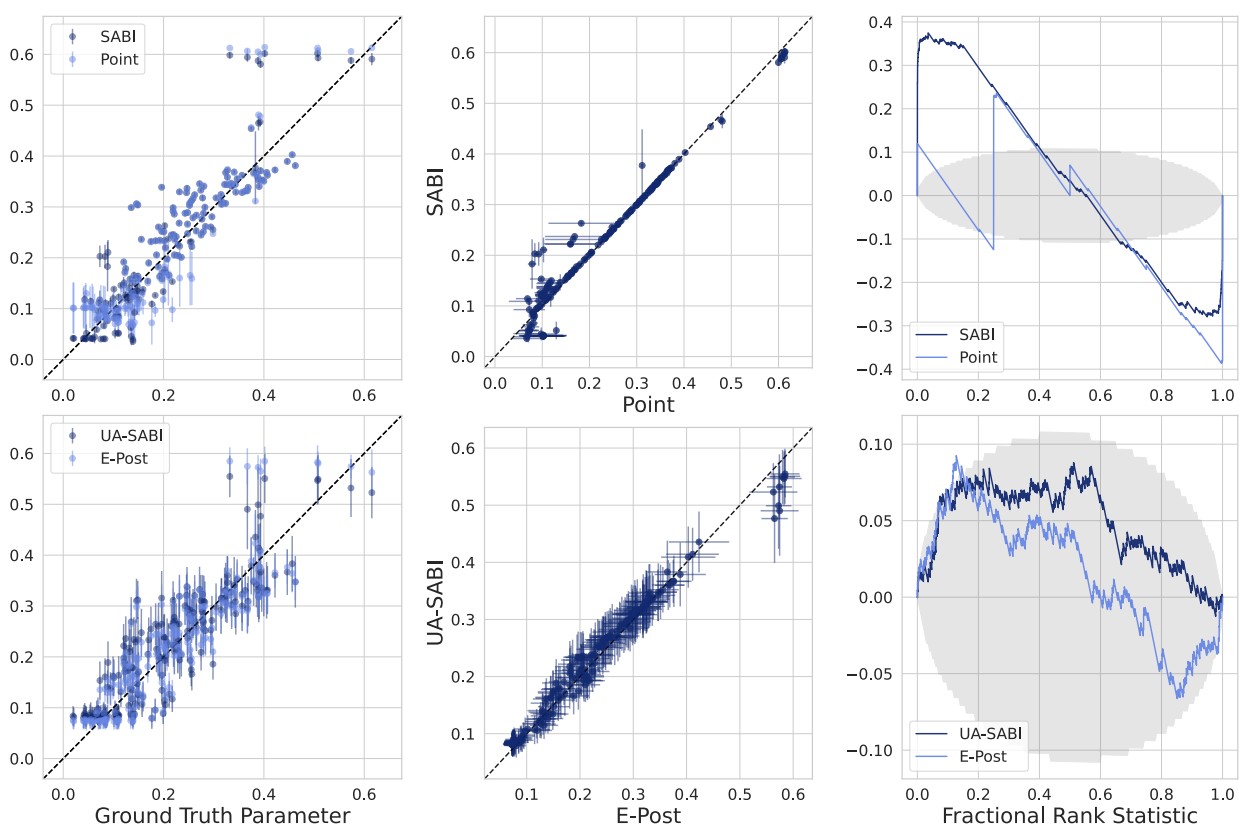

Figure 14: $CO_2$ recovery plots comparing to ground truth and MCMC full reference solution, ECDF difference plots (from left to right) for 200 ground truth samples: SABI vs. Point (top) and UA-SABI vs. E-Post (bottom). For ECDF difference plots, empirical ranks are shown in blue, 95% confidence bands assuming calibration are shown in grey.

# F   Additional Results for Case Study 3: MICP Model

## F.1   Illustration

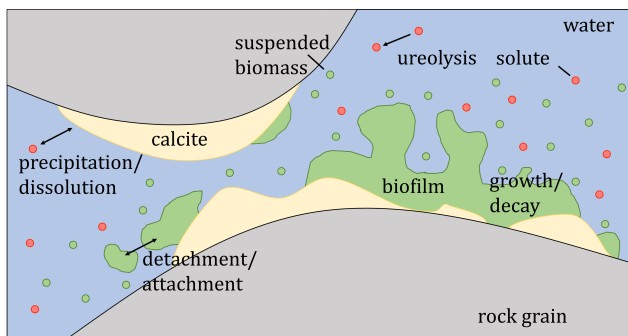

Figure 15: Schematic illustration of relevant processes during MICP. Figure adapted from Scheurer et al. (2021), licensed under CC BY 4.0.

### F.2  Parameter Posterior with Original Priors

Performing inference with the original prior for $k_{ub}$ leads, regardless of the inference method, to point-like posteriors at the upper boundary of the prior. The reason is the same for all methods: the maximal value of $k_{ub}$ yields the highest output values. Since the measurement data are always higher than the surrogate output under the original prior, the posterior collapses to a point-like estimate at the maximum admissible value of $k_{ub}$, as exemplified in Fig. 9.

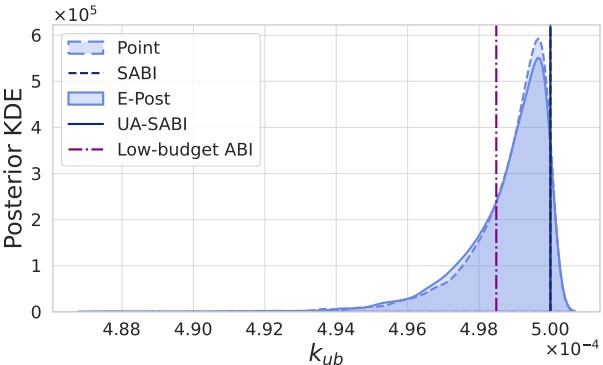

Figure 16: KDEs of 4,000 parameter posterior samples for $k_{ub}$ of the MICP model obtained with low-budget ABI, Point, E-Post, SABI and UA-SABI using the original parameter prior.

### F.3  Sensitivity Analysis

We conducted a sensitivity analysis on the surrogate to identify the parameters to which the surrogate output is most sensitive. Sobol indices (Sobol', 1990) decompose the output variance into contributions from each parameter (including their interactions), with the total Sobol index of a parameter quantifying its overall influence on the output variance. Figure 17 presents the distributions of the total Sobol indices (Le Gratiet et al., 2017) for each parameter at each location, noting that each location has its own surrogate. Since the Bayesian PCE yields samples of the coefficients, the corresponding Sobol indices are also obtained in the form of distributions. Figure 17 clearly shows that almost all the variance in the output at all locations is determined by $k_{ub}$.

### F.4  Parameter Posterior for Remaining Parameters

Figure 18 shows the KDEs of 4,000 parameter samples for the 3 remaining parameters, $c_{a,1}, c_{a,2}$, and $\rho_f$, obtained with low-budget ABI, Point, E-Post, SABI, and UA-SABI. Two distinct scenarios emerge: 1) the posterior collapses to a point estimate (often at the boundary of the prior), or 2) the posterior remains close to

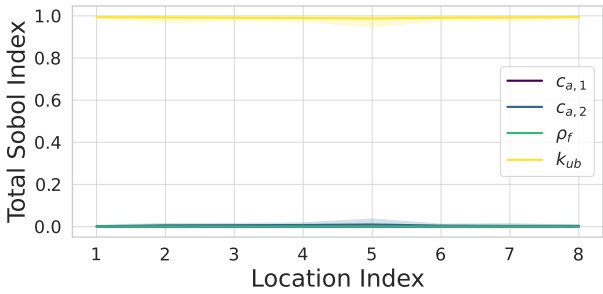

Figure 17: Distribution of total Sobol indices over all locations for the four parameters $c_{a,1}, c_{a,2}, \rho_f, k_{ub}$ of the MICP model.

the prior distribution. Point-like posteriors are produced by low-budget ABI or Point, while surrogate-based methods yield posteriors resembling the prior. Both outcomes can be explained by the lack of sensitivity of the output to these parameters, as shown by the total Sobol indices in Fig. 17. This missing sensitivity makes reliable inference very difficult.

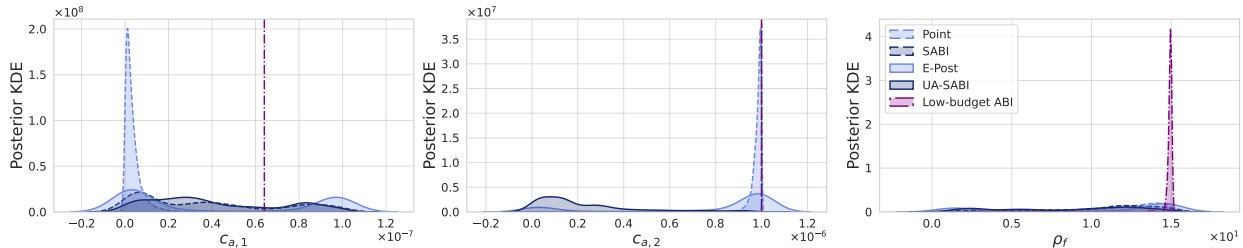

Figure 18: KDEs of 4,000 parameter posterior samples for the remaining parameters $c_{a,1}, c_{a,2}$, and $\rho_f$ of the MICP model obtained with low-budget ABI, Point, E-Post, SABI, and UA-SABI.

### F.5   Runtime Comparison

To compare runtimes as in case studies 1 and 2, inference must be performed on multiple datasets. Since only one dataset of real measurements is available, we conducted the runtime analysis on synthetic datasets. The experiments were also run on a computing cluster with two AMD EPYC 7551 CPUs (totaling 64 physical cores) to speed up E-Post through parallelization.

Figure 19 shows the measured runtimes of UA-SABI and E-Post for $2, 3, 4, 5$ inference runs, with E-Post parallelized across 16 cores. We observe a break-even point between 3 and 4 runs, meaning that UA-SABI becomes advantageous starting from 4 runs. Despite E-Post leveraging more cores, the training costs of UA-SABI are amortized earlier when the model is more expensive.

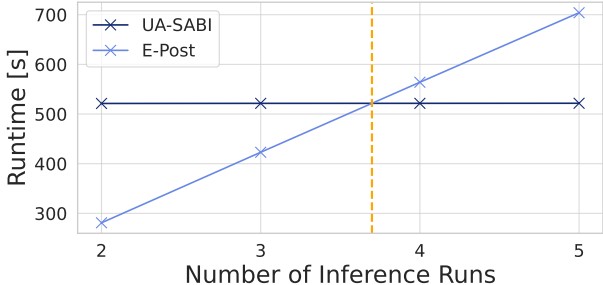

Figure 19: Comparison of runtimes for MICP for {2,3,4,5} runs for UA-SABI (training and inference) and E-Post (inference) with break-even point.

