# OpenReview forum: "Uncertainty-Aware Surrogate-based Amortized Bayesian Inference for Computationally Expensive Models"
_TMLR — Accepted by TMLR_

### Review · Reviewer_BKxS · 2025-11-05

**Summary Of Contributions:**

**Summary**: The paper introduces an amortized Bayesian inference method that explicitly quantifies and propagates surrogate uncertainty into amortized posterior inference, claiming asymptotic equivalence relative to surrogate-marginalizing MCMC (E-post), and empirically shows calibrated uncertainty and amortization benefits on three low-dimensional examples.
Overall, I find that the paper makes a timely and relevant methodological contribution to the simulation-based inference domain and supplements recent work (E-post) by explicitly qualifying and propagating surrogate uncertainty but now using ABI for the final inference. The paper addresses the same basic problem as E-post which relies on MCMC for the final inference step, but the proposal to use ABI appears to be a new direction that provides better scalability than E-post on the inference side.


**Strengths**
  - The scope is clear and sufficiently aligned with the actual content with claims largely supported by evidence (a few unexplained differences between E-post vs. UA-SBI posteriors).
  - Addresses an important practical challenge of conducting Bayesian inference when simulators are expensive.
   - Method-wise I find the combination of surrogate uncertainty with amortized inference sufficiently new/unexplored to be of interest to the community and it compliment the recent E-post method.
  - The methodology is clear with training setup and inference workflow well-defined and the  pseudocode and actual source code aiding reproducibility and impact.
  - Empirical evidence on three problems using relevant ECDF-difference/rank shows a clear advantage over methods not accounting for surrogate uncertainty. The results qualitatively match the E-post method albeit with slight differences, but trades in training cost for reduced inference cost.

**Weaknesses / Limitations**
  - ***Convergence of the UA-SABI posterior to E-post***. A key claim in the paper is that E-post and UA-SABI “targets” the same posterior, but most if not all of the empirical results (4b, 9, 10, E3) shows that there is a small but quantitative difference between the posteriors which is not adequately explained as far as I can tell.
  - ***Possibly too limited scope?*** The authors deliberately and clearly outscope many of the detailed elements/studies, questions and experiments I’d normally expect to see in a method paper of this sort, including:
    - Model assumptions are not really explored, leaving doubts about the method’s robustness to model choices, misspecification and general practicality.
    - Only low-dimensional problems are explored (a deliberate choice) leaving the performance on high-dimensional/complex posteriors unknown.
    - ....and similar low-level experiments that would provide insights into the important elements in the setup; however, I’d be interested in the other reviewers’ and the action editors view on this before coming to a conclusion on whether additional experiments need to be included for publication at TMLR.
  - ***Other*** less critical limitations are addressed with suggestion below.

**Audience:**

Yes

**Audience Explanation:**

Yes, people in the SABI community that worries about reliable posteriors would be interested in this.

**Broader Impact Concerns:**

No concerns

**Claims And Evidence:**

Yes

**Claims Explanation:**

The claims are ***mostly*** supported by evidence but I'd like to see more explanation/discussion of the gap between the UA-SABI and E-post posterior (and ideally a few more optional experiments) and links to the theoretical claim. See Requested Changes for details.

**Requested Changes:**

Note: items listed with **suggest/consider** are not things that would prevent me from potentially recommend acceptance;  **strongly suggest** or **please fix** indicate things that require a response and/or correction.



  -  ***Overview figure***: Figure 1 goes some way in communicating the very high-level concept; however, I **strongly suggest** extending Figure 1 or adding another figure that clearly visualizes the details in a coherent way (perhaps a graphical model) making it easier to appreciate the many elements and various distributions.
  -  ***Objective/loss ( Eq.(2) ) and Algorithm 1***:
     -	The usual SABI loss is stated very briefly in eq. 2. I’d **suggest** providing a bit more context e.g. perhaps simply writing out the derivation in a few more details.
     -	Eq. 2.: Please verify if the expectation should be with respect to p(y) – I believe it to be with respect to p(x,y)?
     - Algorithm 1, line 15 (in the for-loop over N_B observations) mentions that the loss is computed from eq 2; however, this is already a sum over NB terms - **please correct** this.
  - ***Proposition 1***: The wording (i.e. “targets”) seems rather informal. I’d **strongly suggest**; a) improving clarity to more formally state the proposition as an asymptotic convergence result if possible, and b) explicitly listing the corresponding assumptions (expressive estimators, infinite data etc.).
  - ***Discrepancy between E-post and UA-SABI*** As mentioned, I believe the results shown in fig. 4, 9 and 10 (and results in E.3) deserve some explanation/discussion given the key claim that UA-SABI “targets" the same posterior as E-post. I’d **strongly suggest** contrasting the E-post and UA-SABI posteriors and discussing how that gap relates to proposition 1 (see also suggestions for further experiments), i.e. what explains the gap (finite setting, limited expressiveness etc.)? Ideally, I’d **suggest** supporting the discussion by a small empirical study on the convergence of the UA-SABI posterior to the E-post posterior to further back up the claims; however, I’ll not insist on this.
  - ***Supporting experiments*** As previously mentioned, there are several detailed technical experiments I would like to see included. I appreciate that the paper outscoped many of these to provide high-level evidence that the method provides a benefit. Regardless, I’d **suggest** including demonstration/results on at least synthetic high-dimensional problem(s), impact of model choices including misspecification, impact of surrogate model expressiveness and training set size, but I am in doubt whether these should be optional or required, so I will not insist on it at this point.


**Minor comments***:
  - Eq 12: Strictly speaking, I believe this should be $\mathcal{N}( y \mid … )$ instead of $\mathcal{N}( M(x,w) | … )$ - if so, **please correct** it.
  - The code (at least the readme.md) has traces of a NeurIPS submission, I’d **suggest** correcting it before publishing.

---

### Review · Reviewer_Cuvo · 2025-11-09

**Summary Of Contributions:**

The paper introduces uncertainty‑aware surrogate‑based amortised Bayesian inference to enable amortised posterior inference when simulators are too costly to generate sufficient data for training neural posterior estimators. UA‑SABI learns a surrogate from a small set of high‑fidelity simulations and trains the NPE on surrogate‑generated data while propagating both epistemic and approximation uncertainty through training and inference. The authors prove asymptotic equivalence to marginalising over the surrogate posterior (E-Post) and demonstrate, on a toy problem and two real-world case studies that UA-SABI attains accurate posteriors with fewer simulator calls and lower inference cost than MCMC-based alternatives.

Strengths: Clear and well-written, core contribution convincingly demonstrated, interesting real-data examples

Weaknesses: not highly novel - more a (useful) incremental combination and implementation of established ideas

**Audience:**

Yes

**Audience Explanation:**

Yes. The paper covers amortised Bayesian inference with computationally expensive simulators, a common constraint in SBI, and will be of interest to TMLR readers interested in ABI/SBI (e.g., I found this interesting) as well as practitioners in simulation-heavy domains.

**Claims And Evidence:**

Yes

**Claims Explanation:**

Yes each claim is supported: UA-SABI’s uncertainty-aware training and target equality with E-Post are formalised, double amortisation is demonstrated. Evidence includes calibrated recovery/SBC plots and posterior predictives across some compelling case studies with SABI and low-budget ABI consistently overconfident justifying the UA-SABI approach.

**Requested Changes:**

Requested change:

Reduce claims: “we eliminate the training error that arises from insufficient data during ABI training” -> “reduce the training error”.

Other notes (won’t impact acceptance, your discretion to respond to / change in the paper):

While surrogate construction is not the focus here, could there be any further brief guidance on when the use of the surrogate tends to work versus when it is likely to struggle? My main personal reservation if I were to use this myself is just concern if key features of the simulator will be missed by using a surrogate with so few simulations - but I think the case studies address this pretty well (I am surprised and impressed at the decent inference with such few simulations).

Is it possible to contextualise this error modelling approach and its relevance to model misspecification more broadly? Would the error model here also have benefits for simulator misspecification?

When surrogate uncertainty is large, do you ever obtain outputs or parameter regions that violate basic physical bounds? Might this be an issue with the surrogate approach?

---

### Review · Reviewer_MQqn · 2025-11-28

**Summary Of Contributions:**

This paper introduces a simple but seemingly effective uncertainty-aware extension of surrogate-based amortised Bayesian inference (SABI), following the same principles introduced previously by Reiser et al. (2025) for expensive sampling-based inference (MCMC). Like the latter, the proposed method explicitly propagates the approximation uncertainty from the surrogate model, while its amortised nature trades off a one-time training cost for extremely efficient and well calibrated inference.

**Additional Comments:**

- [Sec. 2.2] It could be helpful to give readers some pointers (specific changes, and/or references) for how to account for heteroscedastic (non-i.i.d.) approximation error, even if it's not demonstrated in this paper.
- Consider using more diverse colours and/or markers in the comparison plots, as the shades of blue are hard to distinguish.
- [Sec. 4] Optionally, some illustrations and/or plots for the problem settings in the case studies could make them much easier to follow.
- [Sec. 4.4.2] While marginally interesting on its own, the exploration of prior misspecification of the surrogate is a distraction from the main focus of the experiment. Moving it to the appendix might help streamline the narrative.

**Audience:**

Yes

**Audience Explanation:**

While methodologically incremental, the main contributions of this work are in unlocking cheap approximate inference from few expensive simulations with quantifiable, calibrated uncertainty. Therefore, I believe it could be of interest to methods researchers working on Bayesian inference or to applied researchers in complex domains that often rely on expensive simulators (e.g. geosciences, materials, computational biology).

**Claims And Evidence:**

Yes

**Claims Explanation:**

The method is evaluated on (1) a fully synthetic, univariate, single-parameter task, (2) a real-world-inspired virtual benchmark with 3 target parameters, and (3) a 4-parameter experiment with real measurements. The experimental setup follows the precedent from Reiser et al. (2025): these problems are of similar or higher complexity than those originally supporting E-Post, the same class of surrogates is used (PCE), and evaluation follows the same simulation-based calibration (SBC) protocols.

Within the scope of these experiments, the reported results corroborate the claims of calibrated uncertainty propagation and higher efficiency than the MCMC-based E-post for more than a few inference runs.

On the other hand, although the size of the case studies may be representative of many important real-world problems, higher-dimensional problems were not deliberated in sufficient detail. I understand and agree that running such experiments might be out of scope, though I believe a deeper discussion is warranted about the expected behaviour, challenges, and implications of scaling UA-SABI to higher dimensions.

**Requested Changes:**

1. Please elaborate on the expected behaviour, challenges, and implications of scaling UA-SABI to higher dimensions. This will help readers judge its wider applicability and consider future extensions if appropriate.
2. As a partly unfamiliar reader, I found parts of the text rather dry and not self-contained:
    1. [Introduction] The motivations for the work were not very concrete, and would benefit from some examples. What kinds of simulation problems are we considering? How expensive are they to run and why? What kinds of parameters are we trying to infer, from what kinds of observations?
    2. [Sec. 4.1] As the SBC analysis is the main evaluation, it would be really helpful to explain its principles and interpretation in some more detail.
3. Other parts also need clarification:
    1. [Sec. 2.2] The relationship between $\mathbf{y}$, $\tilde{\mathbf{y}}$, and $\tilde{\mathbf{y}}_e$ was confusing, especially given that the simulated $\mathbf{y}^{(i)}$ is used in eq. (6) in place of the error-adjusted surrogate $\tilde{\mathbf{y}}_e$ from eq. (5). (Using $e$ subscript instead of $\epsilon$ due to a rendering glitch.)
    2. [Sec. 2.1, 4.2.1] It was not clear whether the summary network $S_\theta$ ingests a single pair of inputs-outputs, as suggested in the equations and pseudocode, or a set thereof, implied by its DeepSet implementation.

---

### Decision · Action_Editor_WY4k · 2026-01-07

**Recommendation:** Accept as is

**Audience:**

Yes

**Audience Explanation:**

All reviewers agree that the problem of reducing computational cost for amortized inference in expensive simulators, and the authors' proposed solution, are of interest to a sufficiently large part of the TMLR readership.

**Claims And Evidence:**

Yes

**Claims Explanation:**

All three reviewers recommend acceptance, highlighting that while the submission has certain limitations, e.g., its restriction to low-dimensional settings, its surrogate proposal is a relevant contribution whose claims are well evaluated in a clearly written paper.